

# Investigating Carbon and Nitrogen Conservation in Reported CMIP6 Earth System Model Data

Gang Tang[1,2], Zebedee Nicholls[1,3,4], Chris Jones[5,6], Thomas Gasser[3], Alexander Norton[7], Tilo Ziehn[8], Alejandro Romero-Prieto[9], Malte Meinshausen[1,4]

[1]School of Geography, Earth and Atmospheric Sciences, The University of Melbourne, Melbourne, Australia
[2]Max Planck Institute for Biogeochemistry, Jena, Germany
[3]International Institute for Applied Systems Analysis (IIASA), Laxenburg, Austria
[4]Climate Resource, Fitzroy North, Australia
[5]Met Office Hadley Centre, Exeter, United Kingdom
[6]School of Geographical Sciences, University of Bristol, Bristol, United Kingdom
[7]Research School of Biology, Australian National University, Canberra, Australia
[8]CSIRO Environment, Aspendale, Australia
[9]School of Earth and Environment, University of Leeds, Leeds, United Kingdom

*Correspondence to*: Gang Tang (gang.tang.au@gmail.com; gang.tang@student.unimelb.edu.au; tgang@bgc-jena.mpg.de)

**Abstract.** Reliable, robust, and consistent data are essential foundations for analysis of carbon cycle feedbacks. Here, we consider the data from multiple Earth System Models (ESMs) participating in the Coupled Model Intercomparison Project Phase 6 (CMIP6). We identify a mass conservation issue in the reported carbon and nitrogen data, with a few exceptions for specific models and reporting levels. The accumulated mass imbalance in the reported data can amount to hundreds of
gigatons of carbon or nitrogen by the end of the simulated period, largely exceeding the total carbon/nitrogen pool size changes over the same period. Nitrogen mass imbalance is evident across all reported organic and inorganic pools, with mineral nitrogen exhibiting the most significant cumulative mass imbalance. Due to a lack of details in the reported data, we cannot uniquely identify the cause of this imbalance. However, we postulate that the carbon cycle imbalance in the reported data primarily stems from missing fluxes in the reported data and the inconsistency between the reported data and the
definitions provided by the C4MIP protocol (e.g., land use and fire emissions), rather than from an underlying mass conservation issue in the models themselves. Our findings suggest that future CMIP reporting protocols should consider incorporating mass conservation into their data validation processes so that such issues are caught before users have to deal with them, rather than forcing all users to handle this issue in their own way. In addition, attention from model groups to the detailed diagnostic request and definitions, along with their own quality control will also help to avoid such issues in future.
Given that CMIP6 data is no longer being reported, we recommend that data users that rely on a closed carbon/nitrogen cycle address potential flux imbalances by using the workarounds provided in this study.



## 1 Introduction

The Coupled Model Intercomparison Project (Phase 6 - the latest version, CMIP6) is a collaborative effort among
international climate research communities and plays a fundamental role in advancing our understanding of past, present, and future climate(Meehl et al., 2020; Tokarska et al., 2020; Zelinka et al., 2020). CMIP protocols coordinate and standardize climate model experiments, establish consistent requirements for model outputs, and define protocols for model evaluations (Eyring et al., 2016). The Earth System Model (ESM) outputs serve as a primary model data resource for climate scientists worldwide, supporting a wide range of analyses (such as (Nicholls et al., 2022; Turnock et al., 2020; Stouffer et al.,
2017)) and forming the scientific basis for assessments such as those conducted by the Intergovernmental Panel on Climate Change (IPCC) to inform global climate policy and decision-making (IPCC, 2021a; Arias et al., 2021). Ensuring the reliability, reproducibility, and robustness of the published data is of paramount importance for advancing our understanding of the climate system.

The Coupled Climate-Carbon Cycle Model Intercomparison Project (C4MIP) is one of the CMIP6-Endorsed MIPs focusing on the design, documentation, and analysis of carbon cycle feedbacks and interactions in climate simulations (Friedlingstein et al., 2006; Jones et al., 2016). The interactions between the carbon cycle and atmospheric $CO_2$ concentration, as well as physical climate, are some of the most important feedbacks in the earth system (Friedlingstein et al., 2006; Arora et al., 2013; Arora et al., 2020). The nitrogen cycle also plays significant roles in the carbon cycle feedbacks (Zaehle and
Dalmonech, 2011; Zaehle et al., 2015; Lebauer and Treseder, 2008; Schulte-Uebbing and De Vries, 2018). Accurate and robust estimates of carbon cycle feedbacks rely on careful analysis of model output. The model output is one line of evidence, but is generally of high importance for the question of carbon cycle feedbacks given that there is, to date, little observational evidence with which to constrain these nonlinear feedback processes. Accurate and robust estimates of carbon cycle feedbacks are also highly desirable for developing climate mitigation and adaptation strategies. The need for robust
analysis is why C4MIP made considerable effort to define precise reporting instructions and variable definitions in a way that can be applied to all models (Fig. 1). Tier-1 variables are expected from all models, as is carbon conservation in the reported output (Jones et al., 2016). However, the conservation of carbon and nitrogen in the reported output has not, to date, been assessed.





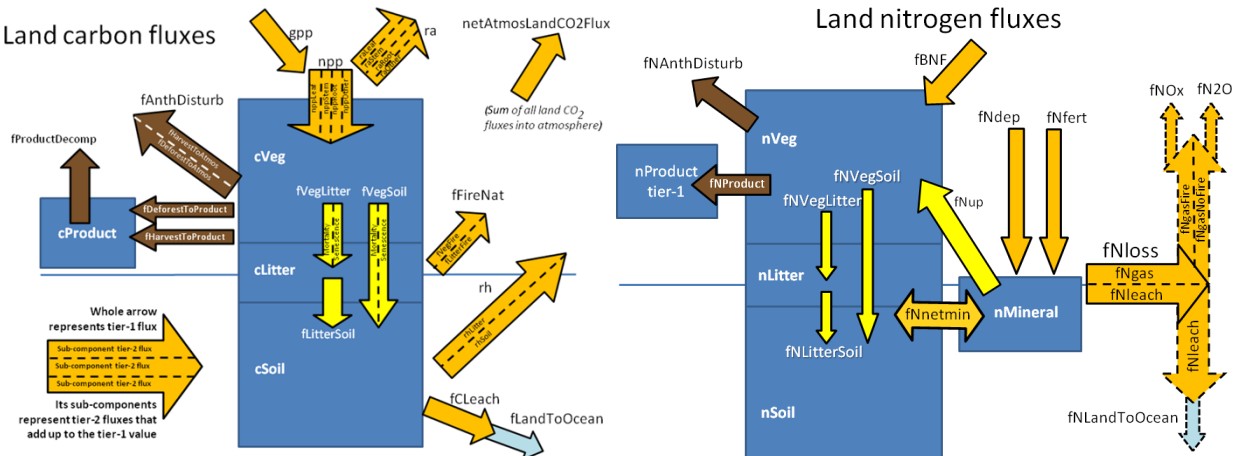

**Figure 1. Land carbon-nitrogen pools and fluxes requested by the C4MIP protocol. This figure is reproduced from C4MIP protocol (Jones et al., 2016) under the Creative Commons Attribution 3.0 License.**

The "data request" in CMIP defines specific requirements for model outputs (variables) when submitting experiment results (Juckes et al., 2020). As one of the most complex elements in the infrastructure, the CMIP6 data request encompasses thousands of variables from the various CMIP experiments (Eyring et al., 2016). Despite detailed standardizations and requirements in variable definitions and output formats, the large number of variables present challenges for data quality control for both modelling groups and Earth System Grid Federation (ESGF) (Petrie et al., 2021). The data quality control issues lead to challenges for data processing and analyses for research communities who use the data, such as end-users who require adjustment of spatiotemporal resolution for various analytical purposes (e.g., estimating carbon cycle feedbacks). Where issues are present, users will either use data that has problems they are unaware of or, rather than being able to use the data directly, have to apply a number of fixes before performing their own analysis.

In this study, we focus on reported land carbon and nitrogen pools and fluxes. Sections 4 and 5 address mass conservation for land carbon and nitrogen, respectively, including their subpools. Each section begins with an introduction to the theoretical mass conservation equations based on the C4MIP protocol (Fig. 1), followed by an analysis of diagnosed mass conservation issues in the reported CMIP6 carbon-nitrogen cycle data. Section 6 discusses additional issues arising from the CMIP6 data request and C4MIP, which contribute to potential confusions. In Section 7, we examine possible causes for mass conservation discrepancies and their implications. Based on the results, we provide suggestions in Section 8 for relevant stakeholders, including the CMIP data request team, ESM groups, the C4MIP and related MIPs, and CMIP6 data users.

As a general disclaimer for the presented work in this manuscript, we do not claim or suggest that there is a mass balance issue within the models themselves. It is a possible explanation, but our study intends solely to highlight how our analysis of



the reported (not raw/internal model) fluxes and pool sizes reveals inconsistencies. These imbalances may stem from various factors: 1) actual mass conservation issues within the models (although we feel that this is highly unlikely given the effort put into model development), 2) missing or incomplete reporting of fluxes (very likely), 3) the methods used by models to diagnose and report pools and fluxes for CMIP6, e.g., spatial-temporal regridding or time-aggregation between the original model output and the reported monthly data, 4) incorrect reporting of aggregate fluxes (i.e. reporting aggregates that don't

match the definition specified in the data request), which could lead to double-counting or missing part of a flux, and 5) errors in our data post-processing, e.g., forming global sums and averages. We hope this manuscript will help shed light on these issues, aiding the community in achieving a unified understanding of definitions, reporting protocols, diagnostic recipes and ultimately providing a cleaner, easier to use set of data in CMIP7 to ensure that the remarkable progress made by modeling teams in recent decades is not lost in data reporting issues (Arora et al., 2020; Jones et al., 2016; Arora et al., 2013;

Friedlingstein et al., 2006).

## 2 Data collection and processing

For the purpose of mass conservation validation, the selection of ESMs and experiments is based on the following criteria: 1) The model must provide a relatively complete set of carbon cycle variables requested by C4MIP. This completeness primarily considers the availability of carbon pool sizes, key fluxes such as 'net primary production' (npp) and

'heterotrophic respiration' (rh), and cross-verification fluxes, for instance, 'net biosphere production' (nbp) and netAtmosToLandCO2Flux. 2) The model must have completed three representative experiments: 1pctCO2, historical, and ssp585. Based on these criteria, fifteen sets of model runs (see "variant_label" in CMIP6 global attributes[1]) from twelve ESMs at nine institutes are included in this study: ACCESS-ESM1-5 (CSIRO, r1i1p1f1), CESM2-WACCM (NCAR, r1i1p1f1), CMCC-CM2-SR5 (CMCC, r1i1p1f1), CMCC-ESM2 (CMCC, r1i1p1f1), CanESM5-1 (CCCma, r1i1p1f1,

r1i1p2f1), CanESM5 (CCCma, r1i1p1f1, r1i1p2f1, r2i1p1f1), MIROC-ES2L (MIROC, r1i1p1f2), MPI-ESM1-2-LR (MPI-M, r1i1p1f1), NorESM2-LM (NCC, r1i1p1f1), NorESM2-MM (NCC, r1i1p1f1), TaiESM1 (AS-RCEC, r1i1p1f1), and UKESM1-0-LL (MOHC, r1i1p1f2). For the same model with different "variant_label", we included only the distinct part - other than "r1i1p1f1" - in the model label for conciseness. For instance, CanESM5 was labeled CanESM5, CanESM5_p2, and CanESM5_r2 for its r1i1p1f1, r1i1p2f1, and r2i1p1f1 runs, respectively.


All variables from the C4MIP protocol (Fig. 1 and CMIP6 data request[2]) from the selected models and experiments were collected based on their availability in ESGF[3]. To ensure consistent data post-processing, the most frequently used monthly data with the model's native grid (with "gn" as "grid_label", see CMIP6 global attributes[1]) was collected and processed into

---

[1] https://goo.gl/v1drZl, last access 25 June 2023
[2] https://clipc-services.ceda.ac.uk/dreq/index.html, last access 25 June 2023
[3] https://esgf.llnl.gov/, last access 25 June 2023

global-mean, annual-mean values. Fig. A1 shows an overview of the data availability for the selected ESMs and
experiments.

As a first step, we used the model-specific grid area ('areacella') and land area fraction ('sftlf') to determine global land pool
sizes and fluxes. Subsequently, the monthly global pool sizes/fluxes were weighted using the model-specific calendar to
calculate their annual means. While the translation from monthly to annual values is not strictly necessary for the mass
conservation test, it aids in clarity and visualization.

## 3 Methods

We define mass conservation in the following way: Mass is conserved when the sum of the fluxes entering and leaving a
carbon/nitrogen pool is equal to the change in the size of the pool. This definition is applied to total pools as well as subpools
(i.e. pools that are a component of some larger pool, e.g. the soil pool, which is part of the overall land pool). The definition
is also applied at all points in time (i.e. the conservation should apply at each individual time step, as well as over time).

To verify mass conservation, we performed the following calculations: First, we reconstructed the net flux by summing the
individual influxes and outfluxes for each pool (Fig. 1). Second, we determined the change in the pool size over time. While
the first calculation is straightforward, the second one requires consideration. Both the pool size data and flux data are
temporal mean values (a requirement from the CMIP6 data request[2]). However, this reporting convention means that there is
only one pool size in each month, rather than information about the size of the pool at the beginning and end of the month
(which is what you would need to unambiguously calculate the change in the pool size over the month of interest). As a
result, blindly calculating the difference in pool size between each reported data point will inevitably result in a net flux that
differs from the flux reported in the time series. Considering this, we employed a gradient method (Fornberg, 1988), which
calculates gradients using second order accurate central differences in the interior points and first order accurate one-sides
differences at the boundaries, to derive the change in the pool size from the reported pool size data. It should be noted that
this approach is simply a method to obtain a better idea of the actual differential (example provided in Text A1 and Fig. A2).
It does not fundamentally resolve the discrepancy between differencing discrete data and differentiating continuous data.
Having said this, the numerical errors introduced due to the reporting convention and the method used to calculate the
change in pool size are expected to be minimal.

## 4 Carbon mass conservation from reported data

To facilitate comparison and formulation, the reconstructed net flux derived from component fluxes and the reconstructed
pool size calculated from subpool sizes are denoted with an asterisk ("*") for both carbon and nitrogen. The mass imbalance



discussed in the following sections, pertaining to both carbon and nitrogen, refers to the discrepancy between the reported
flux data and the change in the pool size. The terms "overestimation" and "underestimation" in this paper refer to the
comparative results between the net flux and the pool size change - one being greater than the other or vice versa.

### 4.1 The top-level closure for land carbon: Equations and results

Based on the C4MIP protocol and CMIP6 data request (Fig. 1), the Tier-1 land carbon pools and directional composite
fluxes should have the following relationships.

$$cLand^* = cVeg + cLitter + cSoil + cProduct \tag{1}$$

$$\frac{dcLand^*}{dt} = netAtmosLandCO2Flux - fCLandToOcean \tag{2}$$

Note that: a) we use the sum of the subpools (cLand*) rather than directly using reported cLand in Eq. 2 to ensure
consistency in carbon conservation analysis between total land carbon and subpool carbon (cLand and cLand* differ in some
ESMs; see details in Section 6.1); b) netAtmosLandCO2Flux and nbp (a CMIP5 variable) are interchangeable based on their
definitions (which may also be part of the confusion, see details in Section 6.2); c) fCLeach and fLandToOcean are in
C4MIP protocol but not in the CMIP6 data request, thus not included in Eq. 2 (see details in Section 6.3); and d)
fCLandToOcean is provided only by CESM2-WACCM (Fig. A1) with negligible values (on the order of 1e-9 GtC/yr, not
shown). Therefore, the following results compare nbp, netAtmosLandCO2Flux, and pool size changes derived from pool
size data (dcLand*/dt).

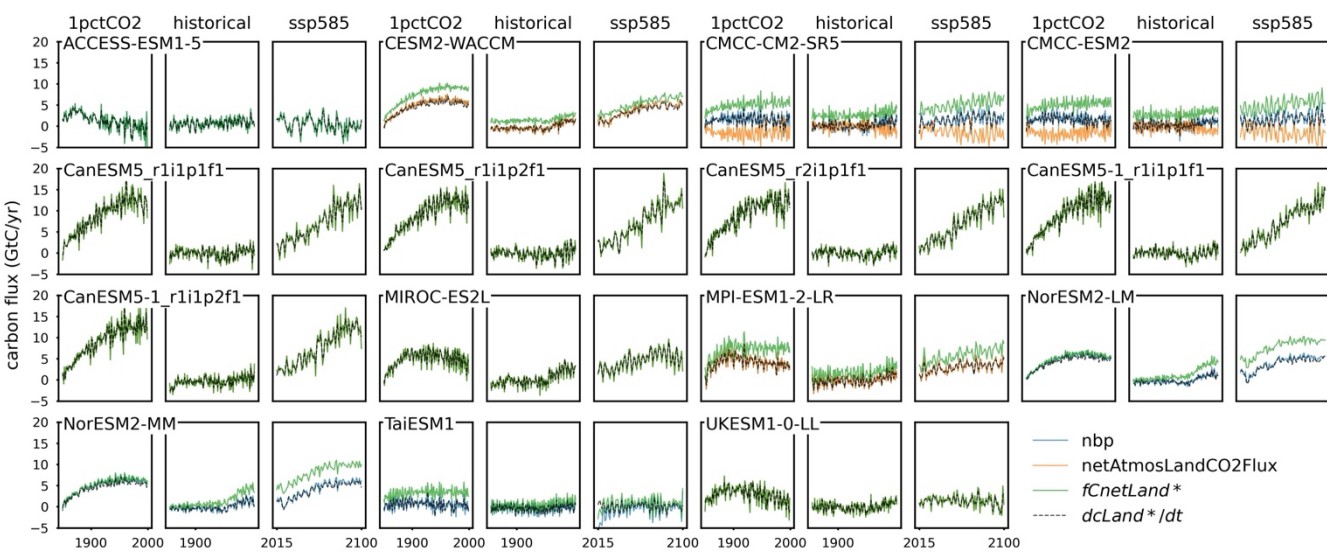

**Figure 2. Comparison of 'net biosphere production' (nbp), netAtmosLandCO2Flux, net land carbon flux (fCnetLand*, reconstructed from flux data), and the change of land carbon pool size over time (dcLand*/dt, derived from pool size data) from CMIP6 ESMs across different scenarios**



The nbp and the dcLand*/dt exhibit the exact relationship as shown in Eq. 2 across all the studied ESMs and experiments
(Fig. 2 and Fig. A3), where the small differences found likely result from the differentiation method applied (see details in
Section 3).

The netAtmosLandCO2Flux and dcLand*/dt are either opposite to each other (the two CMCC models seem to report with a
sign issue, Fig. 2) or exhibit the same trend (all the other models with netAtmosLandCO2Flux provided, Fig. 2). According
to its definition, the netAtmosLandCO2Flux should be identical to nbp, as demonstrated in all ESMs except for the two
CMCC models (Fig. A4). The opposite trends in CMCC models are because of the conflicting description in the C4MIP
protocol and the CMIP6 data request (see details in Section 6.1). Nonetheless, the consistency among nbp,
netAtmosLandCO2Flux (ignoring the sign difference for the CMCC models), and dcLand*/dt indicates three things:

175       a) The models themselves should be carbon mass conserved.

       b) The reliability and robustness of the pool size data and aggregated fluxes (nbp and netAtmosLandCO2Flux).

       c) The modelling teams can report data that conserves mass; hence the root of the issue is very likely in confusion about
how the reporting protocol is meant to be applied.

## 4.2 The closure of land carbon from sum of fluxes: Equations and results

According to the C4MIP protocol and the CMIP6 data requirements (Fig. 1), the Tier-1 land carbon pools and component
fluxes for the net land flux (fCnetLand*) should have the following relationships.

$$fCnetLand^* = npp - rh - fAnthDisturb - fProductDecomp - fFireNat - fCLandToOcean \qquad (3)$$

$$\frac{dcLand^*}{dt} = fCnetLand^* \qquad (4)$$

Note that many ESMs do not report fAnthDisturb (Fig. A1). In such cases, the Tier-2 fluxes fHarvestToAtmos and
fDeforestToAtmos (Fig. 1) are combined to reconstruct it.

Results show that the reconstructed fCnetLand* and dcLand*/dt are well-aligned in ACCESS-ESM1-5, CanESM5-1,
CanESM5, MIROC-ES2L, and UKESM1-0-LL (Fig. 2), with the difference fluctuating around zero and showing no
discernible trends, suggesting that these differences are due to numerical processing issues alone (Fig. A5). In contrast, the
reconstructed fCnetLand* is much higher than the change in the pools in all the other studied ESMs (with the green line
above the black dashed line, Fig. 2). In these models, the disparity between dcLand*/dt and fCnetLand* ranges from 2 to 6
GtC/yr, with an increasing trend in flux differences, particularly in the 1pctCO2 and ssp585 scenarios (Fig. 2 and Fig. A5).
The biased imbalance flux indicates that some component fluxes (outfluxes) in Eq. 3 may be missing from the reported data
or incorrectly reported/double counted.



### 4.3 The significance of cumulative carbon imbalance

Comparing nbp with dcLand*/dt reveals that, although the differences are relatively small and fluctuate around zero (Fig. 2 and Fig. A3), the accumulated mass imbalance is still substantial in certain experiments from some ESMs (Fig. 3). For example, in the 1pctCO2 runs of CESM2-WACCM, NorESM2-LM, and NorESM2-MM, there is an overestimation of approximately 60 GtC (Fig. 3). In contrast, during the historical experiment, nbp and dcLand*/dt are well-aligned, resulting in a negligible accumulated imbalance (Fig. 3). The mass imbalance in the ssp585 run is also small except for the TaiESM1, where the integration of (nbp - dcLand*/dt) reveals an underestimation of 53.8 GtC of carbon pool size change, comparable in magnitude to the overestimation resulting from the (fCnetLand* - dcLand*/dt) integration (48.3 GtC). This indicates that both nbp and fCnetLand* in TaiESM1's ssp585 run are inconsistent with its reported pool size data.

The imbalance flux from the reconstructed fCnetLand* results in an accumulated mass imbalance ranging from 93.7 to 531.8 GtC, 115.1 to 412.9 GtC, and 48.3 to 351.2 GtC for the 1pctCO2, historical, and ssp585 runs, respectively, in models where carbon conservation is not strictly maintained (Fig. 3). These values are substantially higher than the cumulative imbalance from nbp - dcLand*/dt (with a maximum of approximately 60 GtC). By contrast, models that maintain relative carbon conservation (ACCESS-ESM1-5, CanESM5-1, CanESM5, MIROC-ES2L, and UKESM1-0-LL) exhibit much smaller imbalances (<5.5 GtC, Fig. 3), likely reflecting unavoidable numerical errors arising from the differentiation of discrete pool size time series.

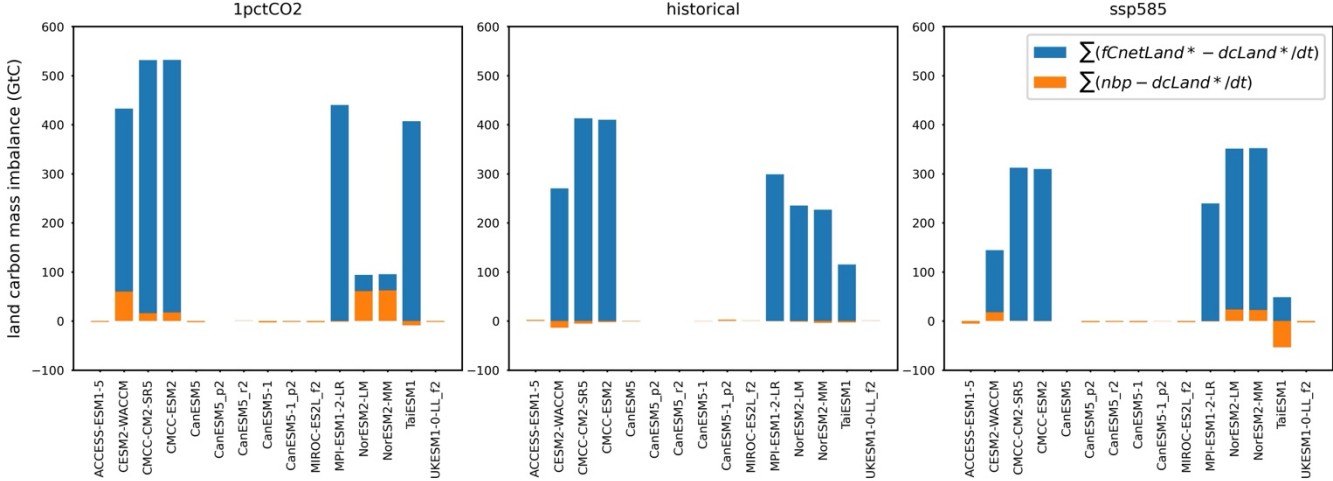

**Figure 3. The carbon mass imbalance from mismatch of net land carbon flux (fCnetLand*, reconstructed from flux data) or 'net biosphere production' (nbp) and the change in land carbon pool size over time (dcLand*/dt, derived from pool size data) for CMIP6 ESMs across different scenarios.**

From the historical period to ssp585, the land carbon storage trends based on pool size data show a decrease from 1850 to 1975, followed by either continued increase or stabilization until 2100, depending on the ESM (Fig. A6). During the



historical period, most ESMs exhibit a decrease in land carbon pool size ranging from 6.8 GtC (CanESM5-1) to 62.6 GtC

(TaiESM1). In contrast, ACCESS-ESM1-5, CMCC-CM2-SR5, CMCC-ESM2, and MIROC-ES2L show accumulation of land carbon ranging from 39.0 to 109.1 GtC. Integration of fCnetLand* indicates a maximum carbon accumulation of 465.4 GtC during the historical period, significantly exceeding the changes observed in land carbon pool sizes (Fig. A6). Additionally, for ESMs exhibiting mass imbalance, none capture the decreasing trend in land carbon before 1975 through integration of fCnetLand* (Fig. A6). Under the ssp585 scenario, land carbon pool sizes increase from 2.4 GtC (TaiESM1) to

600.4 GtC (CanESM5-1). ESMs with mass imbalance show integration of fCnetLand* at least 1.5 times higher than their corresponding changes in pool sizes (Fig. A6). The large discrepancy suggests that ESMs with different fCnetLand* and dcLand*/dt may lead to significant issues in carbon budget (here meaning the internal flows within the carbon cycle, not our remaining carbon budget) assessments. Some of these conservation issues may be solved by considering variables outside the C4MIP protocol (see discussions in Section 7.2).

**4.4 The closure of land carbon in subpools: Equations and results**

The subpools within the land carbon pool should also maintain mass conservation. Note that the mass conservation of subpools involves some Tier-2 fluxes, such as rhLitter and rhSoil (subfluxes for rh, Fig. 1), which are unreported for many ESMs (Fig. A1). Therefore, we combine the litter and soil pools when writing the mass conservation equations for subpools.

$$fCnetVeg^* = npp - fDeforestToProduct - fHarvestToProduct - fAnthDisturb - fVegLitter - fVegSoil - fVegFire \tag{5}$$

$$\frac{dcVeg^*}{dt} = fCnetVeg^* \tag{6}$$

$$fCnetLitterSoil* = fVegLitter + fVegSoil - fLitterFire - rh - fCLandToOcean \tag{7}$$

$$\frac{dcLitterSoil^*}{dt} = fCnetLitterSoil^* \tag{8}$$

$$fCnetProdcut* = fDeforestToProduct + fHarvestToProduct - fProductDecomp \tag{9}$$

$$\frac{dcProduct^*}{dt} = fCnetProduct^* \tag{10}$$

Note that the data availability of Tier-2 variables hinders the mass conservation analysis of subpools (Fig. A1). We thus slightly adjust the calculation of the above equations (see details in Section 6.4).




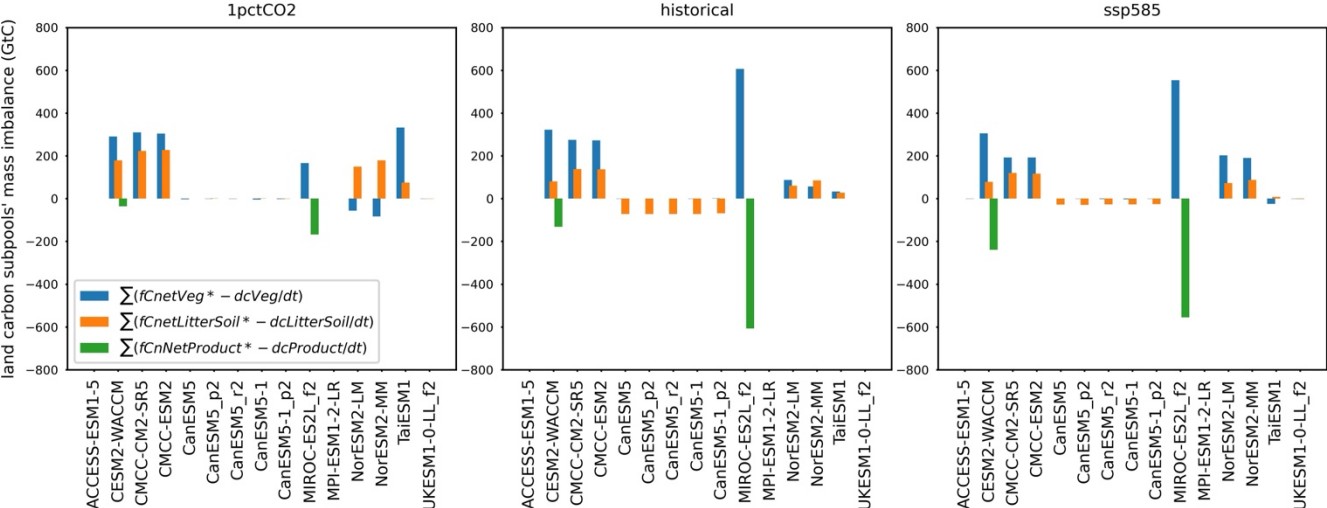

**Figure 4. The carbon mass imbalance from mismatch of net carbon fluxes for vegetation, litter+soil, and product pools**
**(fCnetVeg\*, fCnetLitterSoil\*, and fCnetProduct\*, reconstructed from flux data) and the change in carbon pool size over time for vegetation, litter+soil, and product pools (dcVeg/dt, dcLitterSoil/dt, and dcProduct/dt, derived from pool size data) for CMIP6 ESMs across different scenarios.**

When comparing the reconstructed net flux with the pool size change, it is clear that net flux overestimation or

underestimation occurs across all subpools for nearly all the models (Fig. 4, details in Text A2 and Fig. A7). The cumulative imbalance, calculated from the integration of imbalance flux, shows that the vegetation carbon pool exhibits the highest imbalance in most cases (Fig. 4), with cumulative mass imbalances ranging from ~200 to 600 GtC. Exceptions are found in the 1pctCO2 runs of the two NorESM2 models, where the imbalance from the litter+soil pools surpasses that of the vegetation pools (Fig. 4). MIROC-ES2L, despite achieving overall mass conservation in the total land carbon pool (Fig. 3),

shows an overestimation of the vegetation carbon pool and an underestimation of the product carbon pool across all experiments (Fig. 4). This discrepancy arises from the absence of anthropogenic fluxes fDeforestToProduct and fHarvestToProduct from cVeg to cProduct in MIROC (Fig. 1 and Fig. A1). The CanESM models also conserve total land carbon (Fig. 3); however, their net flux integration reveals an underestimation of the litter+soil carbon pool and an overestimation of the product carbon pool in both the historical and ssp585 runs (Fig. 4). The overestimation in the product

pool is due to the absence of fProductDecomp and cProduct in these experiments (Fig. A1), with only the influx fDeforestToProduct reported (Fig. A1).

**5 Nitrogen mass conservation from reported data**

There is no reference composite flux like nbp or netAtmosLandCO2Flux for the land nitrogen cycle in the C4MIP protocol or CMIP6 data request (Fig. 1). Thus, we start by directly analysing the land nitrogen mass conservation based on the sum of





fluxes. We also analyze the mass conservation from the organic and inorganic nitrogen pools. However, because of a lack of available data (Fig. A1), the further investigation of mass conservation of the organic nitrogen subpools (nVeg, nLitter, nSoil, and nProduct) is not included in this study.

**5.1 The closure of land nitrogen from sum of fluxes: Equations and results**

The Tier-1 land nitrogen pools and component fluxes should have the following relationships (based on the C4MIP protocol,
Fig. 1).

$$nLand^* = nVeg + nLitter + nSoil + nProduct + nMineral \tag{11}$$

$$fNnetLand^* = fBNF - fNAnthDisturb + fNdep + fNfert - fNloss \tag{12}$$

$$\frac{dnLand^*}{dt} = fNnetLand^* \tag{13}$$

The organic and inorganic nitrogen pools should maintain their respective mass conservation as follows.

$$nOrganic^* = nVeg + nLitter + nSoil + nProduct \tag{14}$$

$$fNnetOrganic^* = fBNF - fNAnthDisturb + fNup - fNnetmin \tag{15}$$

$$\frac{dnOrganic^*}{dt} = fNnetOrganic^* \tag{16}$$

$$fNnetMineral^* = fNdep + fNfert + fNnetmin - fNup - fNloss \tag{17}$$

$$\frac{dnMineral}{dt} = fNnetMineral^* \tag{18}$$

Note that: a) similar to the carbon cycle data, we use the sum of subpools (nLand*) rather than the reported nLand for mass conservation analysis; b) because many ESMs do not report fNloss (Fig. A1), we substitute it with Tier-2 variables, 'gaseous nitrogen loss' (fNgas) and 'nitrogen leaching' (fNleach), where necessary.

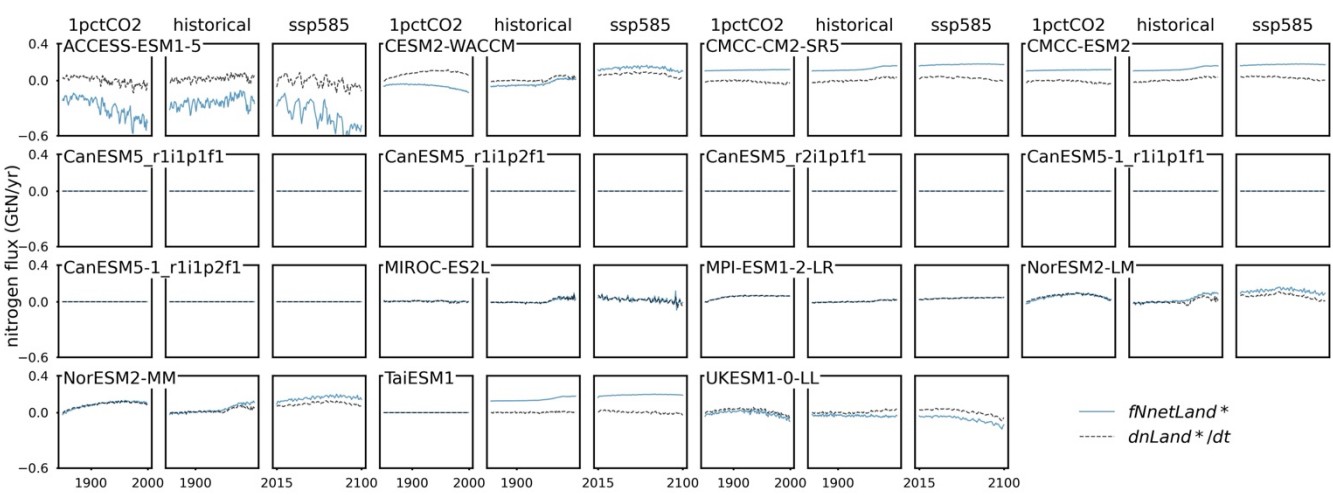





**Figure 5. Comparison of net land nitrogen flux (fNnetLand*, reconstructed from flux data) and the change of land nitrogen pool size over time (dnLand*/dt, derived from pool size data) from CMIP6 ESMs across different scenarios. Note that the CanESM series models do not include a nitrogen cycle.**

In principle, nitrogen mass should be conserved in the organic nitrogen pool, the mineral nitrogen pool, and the total land

nitrogen pool (Eqs. 11-18). However, there is a significant discrepancy between the reconstructed net flux and pool size change over time across nearly all land nitrogen pools in all ESMs and experiments (Fig. 5 and Fig. A8). Notably, only MIROC-ES2L and MPI-ESM1-2-LR show mass conservation from the reconstructed net land nitrogen flux (fNnetLand*) in all three experiments (Fig. 5). The two NorESM models show fNnetLand* matching the land nitrogen pool size change (dnLand*/dt) in their 1pctCO2 runs (Fig. 5).


The gap between the reconstructed net flux and pool size change for the land organic and inorganic nitrogen pools are significantly larger than that for the total land nitrogen pool (Fig. A8, note the larger y-axis scale). For the organic nitrogen pool, the fNnetOrganic* and dnOrganic*/dt only match well in the three experiments from UKESM1-0-LL (Fig. A8). For the inorganic nitrogen pool, while in some models, like the two CMCC models and UKESM1-0-LL, the fNnetMineral* and

dnMineral*/dt time series are very close to each other (Fig. A8), the cumulative imbalance is still significant considering the small mineral nitrogen pool size (Fig. A9).

## 5.2 The significance of cumulative nitrogen imbalance

Regarding the cumulative nitrogen imbalance, in ACCESS-ESM1-5, the fluxes underestimate the change in the total land nitrogen and mineral nitrogen pools, while the fluxes overestimate the change in the land organic nitrogen pool, with a mass

imbalance ranging from 20 to 80 GtN (Fig. 6). In CESM2-WACCM, the reconstructed fNnetLand* underestimates the change in the land nitrogen pool size (dnLand*/dt) for the 1pctCO2 and historical experiments by 20.9 GtN and 7.3 GtN, respectively, while overestimating it by 5.5 GtN in the ssp585 run (Fig. 6). The total land nitrogen changes in CESM2-WACCM for these experimental periods are +11.7, +1.7, and +5.7 GtN (Fig. A9), respectively. This significant imbalance shows that the change in total land nitrogen varies depending on whether it is assessed using pool size or flux data.






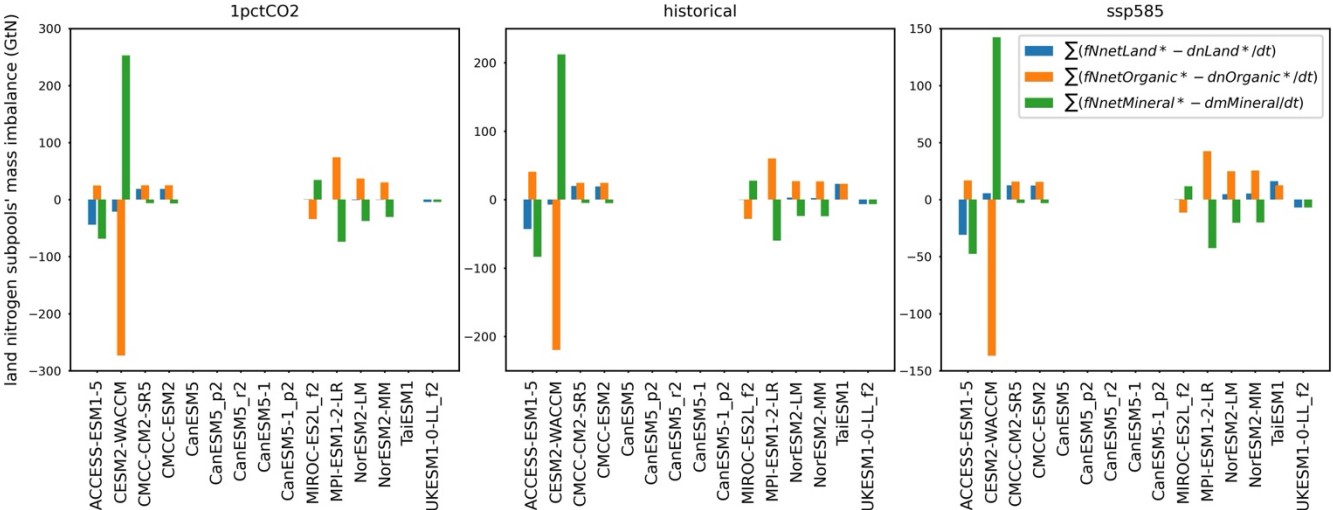

**Figure 6. The nitrogen mass imbalance from mismatch of net nitrogen fluxes for total land, land organic, and land inorganic pools (fNnetLand\*, fNnetOrganic\*, and fNnetMineral\*, calculated from flux data) and the change of nitrogen pool size over time for total land, land organic, and land inorganic pools (dnLand\*/dt, dnOrganic\*/dt, and dnMineral/dt, derived from pool size data) for CMIP6 ESMs across different scenarios. Note that the CanESM series models do not include a nitrogen cycle.**

Examining the mass conservation of the organic and mineral land nitrogen subpools, the net flux in CESM2-WACCM significantly underestimates the organic nitrogen pool size change by 136.9 to 273.6 GtN while overestimating the mineral nitrogen pool size change by 142.4 to 252.7 GtN (Fig. 6). CMCC-CM2-SR5 and CMCC-ESM2 both overestimate the total land nitrogen and organic nitrogen while underestimating the mineral nitrogen, with a mass imbalance of less than 25 GtN (Fig. 6). For CMCC-CM2-SR5, the fNnetLand\* consistently exceeds the land nitrogen pool size change, with overestimation ranging from 12.6 to 19.7 GtN. This imbalance accounts for at least five times the total land nitrogen change for the experimental periods in this model (<2.2 GtN, Fig. 6). MIROC-ES2L, MPI-ESM1-2-LR, NorESM2-LM, and NorESM2-MM are relatively mass-conserved in their total land nitrogen pool (mass imbalance <5 GtN), particularly in their 1pctCO2 runs (mass imbalance <0.8 GtN). However, this conservation is achieved at the expense of simultaneously underestimating the land organic nitrogen pool size while overestimating the mineral nitrogen pool size (for MIROC-ES2L) or vice versa (for the other three). Both the overestimations and underestimations are substantial, ranging from 25 to 75 GtN (Fig. 6). The reconstructed net flux in TaiESM1 shows an overestimation of its land nitrogen pool size in the historical and ssp585 experiments (~25 GtN, with no data provided for its 1pctCO2 run), primarily due to the overestimation of the organic nitrogen pool. In UKESM1-0-LL, the reconstructed net flux overestimates the total land nitrogen by 4.1 GtN in the 1pctCO2 experiment and underestimates it by 6.7 and 7.0 GtN in the historical and ssp585 experiments, respectively. The mass imbalance for UKESM1-0-LL predominantly arises from its organic nitrogen pool.




Based on reported pool size data, the total land nitrogen changes across all studied ESMs during the 1pctCO2, historical, and ssp585 experimental periods range from -2.0 to +13.1 GtN, +0.4 to +3.8 GtN, and -1.7 to +5.5 GtN, respectively (Fig. A9). Changes in the mineral nitrogen pool are substantially smaller, remaining below 0.3 GtN for all experiments and ESMs, with the exception of the 1pctCO2 run in NorESM2-LM (Fig. A9). Comparatively, the cumulative nitrogen imbalance from the reconstructed net flux (Fig. 6) is orders of magnitude larger than the total nitrogen pool size changes over the same periods.

## 6 Other issues

### 6.1 The reported cLand

In many of the ESMs examined in this study, the reported total land carbon (cLand) does not equal the sum of the respective subpools (i.e., the reconstructed cLand*). The discrepancy ranges from 50 to 150 GtC on average during the experimental period. This inconsistency is unexpected, as pool size, being a state variable, should be simple to calculate and report. This hence indicates that modeling teams may face challenges in fully adhering to C4MIP guidelines, an issue that warrants attention as we approach CMIP7. However, without further evidence (higher resolution or more comprehensive datasets), we are unable to analyze the underlying causes in the CMIP6 data.

### 6.2 The definition of nbp and netAtmosLandCO2Flux

The netAtmosLandCO2Flux was a new variable introduced in the CMIP6 phase of C4MIP, defined as the "Net Flux of CO2 Between Atmosphere and Land (Positive into Land) as a Result of All Processes." Many models do not report this flux (Fig. A1). Instead, all models report 'net biosphere production' (nbp), defined as the "Carbon Mass Flux out of Atmosphere Due to Net Biospheric Production on Land," a variable introduced in CMIP5. Based on their definitions, netAtmosLandCO2Flux and nbp appear interchangeable.

However, it should be noted that in the C4MIP paper (Jones et al., 2016), netAtmosLandCO2Flux is illustrated as directed from land to atmosphere (Fig. 1) and described as the "total flux of CO2 from land to atmosphere," which conflicts with its definition in the CMIP6 data request and may be misleading (e.g., the opposite signs for netAtmosLandCO2Flux and nbp observed in the CMCC models, Fig. 2). Resolving these issues may help reduce confusion and improve the consistency of reported data.

### 6.3 The changes in fCLeach and fLandToOcean from C4MIP protocol to CMIP6 data request

The original C4MIP design includes a 'leaching carbon flux' (fCLeach) as a Tier-1 variable (Fig. 1). However, this flux is not in the CMIP6 data request (and therefore ESGF) and is thus excluded from Eq. 2. Additionally, fLandToOcean in the C4MIP protocol is renamed to fCLandToOcean in the CMIP6 data request. These changes introduce potential confusion for data users and suggest that a living variable definition and protocol may be essential for providing groups with clarity, given



the slow moving nature of paper publication (which makes papers a source that quickly becomes outdated). Living
data/protocol papers may provide another solution which could be updated quickly enough to avoid becoming quickly out of
date.

## 6.4 The adjustments for conservation calculations due to limited data availability

For carbon conservation in vegetation and litter-soil pools, consideration of the Tier-2 component fluxes (fVegFire and
fLitterFire) for fFireNat is unavoidable (Eq. 5 and Eq. 7). However, fFireNat itself is reported only in TaiESM1 and the two
NorESM models (Fig. A1), and component fluxes are even rarer (only CESM2-WACCM reports fVegFire, and none of the
studied models report fLitterFire, Fig. A1). Consequently, in calculating fCnetVeg* and fCnetLitterSoil, we replace
fVegFire in Eq. 5 with fFireNat and omit fLitterFire from Eq. 7. While this adjustment is not ideal, it is the only feasible
approach given the limited data availability.

It is common for ESMs to report Tier-2 variables while omitting Tier-1 variables for certain fluxes (e.g., fHarvestToAtmos
and fDeforestToAtmos vs. fAnthDisturb; fNgas and fNleach vs. fNloss, Fig. 1 and Fig. A1). This does not pose a major
issue, as we can reconstruct the composite variables by combining the component variables, as done in the conservation
calculations. However, this suggests that the C4MIP or CMIP6 data request could consider upgrading the priority of some
component variables and potentially removing certain composite variables.

## 7 Discussions

### 7. 1 Top level implications

First, it is important to note that we cannot directly determine whether there is a true "mass conservation issue" in the
physics of the ESMs based on the published data. From a modeller's perspective, such an issue seems unlikely. Furthermore,
we cannot directly verify discrepancies that may arise from the post-processing of the original ESM output to meet the
CMIP6 data requirements. It is worth noting that CMIP6 requested variables, the most common of which is monthly gridded
data, do not match the original temporal solving frequency in ESMs (time steps also vary within ESMs' sub-modules)
(Thornton et al., 2007; Sokolov et al., 2008; Wiltshire et al., 2021; Lawrence et al., 2019). This means that post-processing
steps like temporal averaging and spatial summing/regridding are inevitable. Numerical errors or other issues may occur
during this data processing. Without thoroughly examining the ESM groups' post-processing steps for CMIP6 data
submission, it is impossible to pinpoint the source of the mass imbalance. Therefore, the following discussion of the mass
conservation issue refers solely to the notable mismatch between the flux data and pool size data submitted to CMIP6/ESGF
by ESMs - based on the C4MIP protocol.



The inconsistency between changes in pool size (derived from pool size data) and the reconstructed net flux (derived from
flux data, as detailed in Sections 4 and 5) is substantial and is observed across various models and experiments for both
carbon and nitrogen, including their subpools (however limited data availability for subpools precludes further discussion
here). This discrepancy warrants careful attention in future data requests and presents challenges for current data users.

The carbon mass conservation analysis reveals that the land carbon pool size data align well with both the nbp and
netAtmosLandCO2Flux (where available, and ignoring sign issues) in most cases (Fig. 2), which enhances the credibility of
the pool size data. The matching of nbp and pool sizes (state variables) also supports the assertion that the ESMs maintain
mass conservation in their raw data.

### 7.2 The inconsistent land carbon pool size change (dcLand*/dt) and net flux (fCnetLand*)

To understand the potential causes of the land carbon conservation issue, we look into the component fluxes in the net flux
(Eq. 3), which include npp, rh, fAnthDisturb, fFireNat, fProductDecomp, and fLandToOcean. For clarity, Table 1 provides a
summary of the land carbon mass conservation status across the models and experiments analyzed.

**Table 1. Land carbon mass conservation based on balance between reported fluxes and changes in pool sizes (Equations 3 and 4)**

| | land carbon mass conservation status | | | | | | | | |
|---|---|---|---|---|---|---|---|---|---|
| Variable processing | C4MIP variables only | | | add fLuc | | | add fFire | | |
| Models | 1pctCO2 | historical | ssp585 | 1pctCO2 | historical | ssp585 | 1pctCO2 | historical | ssp585 |
| ACCESS-ESM1-5 | Y | Y | Y | / | / | / | / | / | / |
| CESM2-WACCM | N | N | N | N (n.d.) | N (-) | N (-) | N (-) | N (+) | N (+) |
| CMCC-CM2-SR5 | N | N | N | N (n.d.) | N (-) | N (-) | N (n.d.) | N (n.d.) | N (n.d.) |
| CMCC-ESM2 | N | N | N | N (n.d.) | N (-) | N (-) | N (-) | N (-) | N (-) |
| CanESM5 | Y | Y | Y | / | / | / | / | / | / |
| CanESM5_p2 | Y | Y | Y | / | / | / | / | / | / |
| CanESM5_r2 | Y | Y | Y | / | / | / | / | / | / |
| CanESM5-1 | Y | Y | Y | / | / | / | / | / | / |
| CanESM5-1_p2 | Y | Y | Y | / | / | / | / | / | / |
| MIROC-ES2L_f2 | Y | Y | Y | / | / | / | / | / | / |



| MPI-ESM1-2-LR | N | N | N | N (-) | N (-) | N (-) | Y | N (-) | N (+) |
| NorESM2-LM | N | N | N | N (-) | N | N (-) | N (n.d.) | N (n.d.) | N (n.d.) |
| NorESM2-MM | N | N | N | N (n.d.) | Y | N (-) | N (n.d.) | / | N (n.d.) |
| TaiESM1 | N | N | N | N (n.d.) | Y | N (+) | Y | / | N (n.d.) |
| UKESM1-0-LL_f2 | Y | Y | Y | / | / | / | / | / | / |

Y: mass conserved (cumulative imbalance <= 5GtC)

N: mass not conserved (cumulative imbalance > 5GtC)

N (-): mass not conserved, but including this variable decreased mass imbalance (make things better)

N (+): mass not conserved, and including this variable increased mass imbalance (make things worse)

N (n.d.): mass not conserved, no data for this variable

/: mass conserved with C4MIP variables so no further processing

**7.2.1 fAnthDisturb vs. fLuc: The land use emissions reported outside C4MIP**

The fAnthDisturb variable, comprising fDeforestToAtmos and fHarvestToAtmos (Fig. 1), is related to land use, land-use change, and forestry (LULUCF) emissions (Houghton et al., 2012; Gasser and Ciais, 2013; Stocker and Joos, 2015; Lawrence et al., 2016; Hurtt et al., 2020). Among the studied ESMs, only UKESM1-0-LL - showing negligible mass imbalance (Fig. 2 and Fig. 3) - reports an increasing fHarvestToAtmos from 1 to approximately 5 GtC/yr during the

historical to ssp585 periods. The historical fAnthDisturb in UKESM1-0-LL aligns with modelled LULUCF emissions from various dynamic global vegetation models (DGVMs, process-based) and bookkeeping models (e.g., OSCAR, used in the Global Carbon Budget program) (Gasser et al., 2020).

Except for UKESM1-0-LL, CanESM5, CanESM5-1 (both mass-conserved in the land carbon pool), and CESM2-WACCM

(not mass-conserved) have reported fDeforestToAtmos (0 - 1.5 GtC/yr). The other ESMs have not provided any data for fAnthDisturb, fDeforestToAtmos, or fHarvestToAtmos (Fig. A1). The definitions of these fluxes are similar to those of gross land-use emissions. Their absence in the historical and SSP585 experiments appears unrealistic and may contribute to the diagnosed carbon mass conservation issue. This is partially corroborated by the mass conservation analysis of the land carbon subpools (Text A2 and Fig. A7), where the vegetation carbon pool - which also has fAnthDisturb (or

fDeforestToAtmos, and fHarvestToAtmos) as an outflux (like the total land carbon pool, Fig. 1) - accounts for most of the land carbon mass imbalance in several ESMs (Fig. 4).

To further investigate the land use fluxes, we analyze the fLuc ("Net Carbon Mass Flux into the Atmosphere Due to Land-Use Change") variable from the CMIP5 data request. With the exception of ACCESS-ESM1-5, CanESM5-1, CanESM5, and





MIROC-ES2L (which are already land carbon mass-conserved, Fig. 3), all other ESMs report fLuc (Fig. A1), with values ~4 GtC/yr for the two NorESM models and <1.5 GtC/yr for most of the others. Since fLuc is not included in the C4MIP protocol (Fig. 1), it does not fit into the equations used for mass conservation calculations (see Section 3). Given its definition, we assume fLuc as an additional outflux from the land carbon pool and recalculate the imbalance accordingly.

NorESM2-MM and TaiESM1 are mass-conserved in their historical runs when fLuc is considered, with carbon mass imbalances dropping from 226.5 and 115.1 GtC to 4.7 and 3.1 GtC, respectively (Table 1 and Fig. A10). The mass imbalance in the NorESM2-LM's ssp585 run also significantly decreases from 351.2 GtC to 22.0 GtC (Fig. A10). Other ESMs with fLuc considered show varying degrees of imbalance reduction (Fig. A10), although the imbalances remain substantial (>55 GtC). The fLuc results further highlight the importance of including land use emissions for mass conservation. However, the discrepancy between the C4MIP protocol and ESM data reporting obscures this issue and may cause confusion.

### 7.2.2 fFireNat vs. fFire: The fire emissions reported outside the C4MIP

NorESM2-LM, NorESM2-MM, and TaiESM1 are the three models that report fFireNat (Fig. A1). However, the mass imbalance in these four models remains substantial (Fig. 2 and Fig. 3). We notice there is a similar CMIP5 requested variable with fFireNat, fFire ("Carbon Mass Flux into Atmosphere Due to CO2 Emission from Fire Excluding Land-Use Change"), which is reported by many ESMs (Fig. A1). Thus, we re-check the mass conservation by replacing fFireNat with fFire, where fFire is available.

It is found the fFire is quite high in the 1pctCO2 experiment for the models reporting this flux (e.g., in the range of 1 to 4 GtC/yr with an increasing trend). Considering fFire in the 1pctCO2 run leads to the mass conservation of MPI-ESM1-2-LR and TaiESM1, with the imbalance of 1.6 and 9.1 GtC, respectively (Table 1 and Fig. A11). The imbalance also drops from 432.7 to 59.7 GtC in CESM2-WACCM and from 531.8 to 207.0 GtC in CMCC-ESM2 (Fig. A11). In the historical and ssp585 experiments, though the consideration of fFire largely decreases the imbalance in CMCC-ESM2, it results in some negative imbalance in CESM2-WACCM and MPI-ESM1-2-LR (i.e., fFire is too large). That indicates there might be double counting when simply replacing fFire with fFireNat, which complicates the analysis.

### 7.2.3 The remaining component fluxes

Except for the fAnthDisturb and fFireNat, there are four other fluxes determining the net land carbon fluxes (Eq. 3): npp, rh, fProductDecomp, and fCLandToOcean. Among them, npp and rh stand out as the major carbon fluxes, extensively studied and analyzed in numerous investigations (Jian et al., 2022; Wei et al., 2022; Zhu et al., 2022; Qiu et al., 2023). Observations contribute to constraining the uncertainty of modelled npp and rh (Haverd et al., 2013; Guenet et al., 2024), thereby enhancing their data credibility.



The flux fProductDecomp is reported by ACCESS-ESM1-5, CESM2-WACCM, CMCC-CM2-SR5, CMCC-ESM2, MIROC-ES2L, MPI-ESM1-2-LR, and UKESM1-0-LL (Fig. A1). Notably, four of these seven ESMs demonstrate land carbon mass conservation (Fig. 3). During the historical to SSP585 period, fProductDecomp increases significantly, reaching up to 5 GtC/yr in CESM2-WACCM and MPI-ESM1-2-LR, and as high as 7 GtC/yr in MIROC-ES2L. This substantial flux underscores its importance in closing the carbon cycle. The product pool mass imbalance observed in MIROC-ES2L highlights the importance of carbon fluxes into the product pool (Fig. 4).

Among the studied ESMs, CESM2-WACCM is the only model that provides fCLandToOcean. As most of the CMIP6 ESMs may not simulate the river transport and we have not found other similar CMIP6 variables representing the carbon from land to ocean, we do not discuss this flux here.

### 7.3 The complexity of carbon mass conservation for subpools

In principle, land carbon subpools should also adhere to mass conservation (Eqs. 5-10). However, the results indicate that most do not, even for those ESMs where total land carbon is mass conserved (Fig. 4 and Fig. A7). Given that fluxes for subpool mass conservation follow specific pathways - such that an outflux for one subpool serves as an influx for the next (e.g., fVegLitter is an outflux for cVeg and an influx for cLitter) - imbalanced subpools present challenges in pinpointing the potentially problematic fluxes. Therefore, it would be unreasonable to attribute the imbalance to any single flux based solely on the reported data. This complexity necessitates a more detailed examination of the underlying processes and how these fluxes are reported to identify sources of mass imbalance in the subpools (see Section 8 below).

### 7.4 Nitrogen mass conservation: An emerging and challenging issue

The conservation of land nitrogen mass presents a more intricate challenge due to the complexities inherent in the nitrogen cycle (Fig. 1). The total land nitrogen, as well as the organic and mineral nitrogen pools, should maintain their respective mass conservation (Eqs. 11-18). However, current CMIP6 data only supports mass conservation in the total land nitrogen pool for MIROC-ES2L, MPI-ESM1-2-LR, NorESM2-LM, and NorESM2-MM, and in the organic nitrogen pool for UKESM1-0-LL (Fig. 6). The imbalance in the reported mineral nitrogen pool is also significant across all the studied CMIP6 ESMs (Fig. 6).

Unlike the carbon cycle, there is no reference nitrogen flux equivalent to nbp or netAtmosLandCO2Flux in the CMIP6 data request and C4MIP design, which complicates the direct validation of both nitrogen pool size data and flux data (Fig. 1). The current fNloss includes fluxes into both the atmosphere (fNgas) and the ocean (fNleach), making it challenging to define a clear reference flux (Fig. 1). Additionally, fNup and fNnetmin link the organic and mineral pools in two directions (Fig. 1), further complicating the attribution of the imbalances.





It is worth noting that only MPI-ESM1-2-LR and UKESM1-0-LL have reported the fNAnthDisturb flux. While MPI-ESM1-2-LR is land nitrogen mass conserved and UKESM1-0-LL is not, this suggests that attributing the total land nitrogen imbalance solely to anthropogenic disturbance may not be appropriate. The lack of a straightforward reference flux and the complex interconnections within the nitrogen cycle highlight the need for more detailed scrutiny to identify and address the sources of nitrogen mass imbalance in these models.

**7.5 The quest for reporting balanced carbon and nitrogen cycles**

The carbon and nitrogen cycle data, together with many climate variables, are frequently used to support analyses with various purposes (IPCC, 2021b; Arora et al., 2020; Thornhill et al., 2021; Canadell et al., 2021; Hermans et al., 2021; Fan et al., 2020). These results are particularly crucial for reduced complexity models (emulators), decision-making, and climate policy (Meinshausen et al., 2022; Koven et al., 2022; Kikstra et al., 2022). Considering the challenges in data storage and 510 distribution, it is reasonable that CMIP requests the "most useful" data rather than universally useful output. There is no "one solution for all" for the CMIP data request, meaning that different research groups require varying levels of post-processing based on their specific purposes. This means that ensuring consistency between the model's original outputs and the processed data is important.

Ensuring mass conservation is crucial for data users, especially those analyzing the comprehensive behaviors of the full carbon-nitrogen cycle, such as reduced complexity modeling groups (Meinshausen et al., 2011a; Meinshausen et al., 2011b; Nicholls et al., 2021; Nicholls et al., 2020). Estimating carbon cycle feedback also relies on published ESM data (Arora et al., 2020; Melnikova et al., 2021). Although the nitrogen cycle is relatively new in ESMs (Davies-Barnard et al., 2022), its significance has been widely discussed and demonstrated in both experimental results and models (Zaehle et al., 2010; 520 Zaehle and Dalmonech, 2011; Zaehle et al., 2014; Schulte-Uebbing and De Vries, 2018). Addressing mass conservation of carbon and nitrogen data, a fundamental requirement for accurate analysis, remains a significant challenge. Achieving mass conservation in reported data would significantly simplify the life of users, facilitating more science done faster.

**8 Suggestions for the path forward**

Based on the preceding discussions, the following practical recommendations are provided for the CMIP data request team, 525 modelling groups, C4MIP and related MIPs (for the future development) and CMIP6 data users, focusing on carbon-nitrogen cycle data.



## 8.1 For the CMIP data request and ESM groups

a) It is suggested that the next generation of the CMIP data request and the ESM groups incorporate mass conservation of reported data as a data validation routine. ESM groups may consider reporting fluxes that are consistent with the pool size changes, i.e. there may need to be a better way to represent the assumptions required to ensure consistency between the sum of the reported fluxes and the change in pool sizes. Since the process from original model outputs to reported data is not fully transparent to data users, any imbalances in the reported data, traceable to the smallest subpools the model can analyze, must be documented by the ESM groups. This documentation should include the destination of the imbalance flux (e.g., atmosphere or ocean) and the underlying causes (model process or technical issues like temporal averaging). ESGF may develop necessary data quality control tools considering mass conservation for data publication.

At first, this may seem a difficult task to meet. However, we would ask a different question. What is the value of sharing data which does not pass the basic test of conserving mass? Given the effort it requires to move and track such large data volumes, basic validation checks seem like the minimum hurdle which should be cleared before the data is deemed worthy of the effort required to push it into the wider ecosystem.

b) Direct communication between the team behind the CMIP data request (based on MIP protocols) and the ESM groups should be strengthened to address data reporting challenges and to inform updates to MIP protocols accordingly. Currently, a disconnect exists between these two - ESMs often find it challenging to fully align with MIP protocols, leading to incomplete data reporting or reported fluxes that do not adhere to the prescribed definitions (as evidenced by the fLuc and fFire). Bridging this gap requires a collaborative effort and should be addressed promptly to improve data quality and make the substantial data sharing effort done by the ESGF worth it.

c) The CMIP data request may consider mandating complete data submission from ESM groups for MIPs (e.g. a core set of required variables like the Tier-1 variables in C4MIP) before publication. In particular, reporting Tier-1 variables should be a prerequisite for reporting Tier-2 variables to prevent confusion among data users. For variables requested by MIPs that are not simulated by ESMs, ESM groups should provide a data availability statement for the variable rather than simply not reporting them. Specifically for C4MIP, the note may indicate how to conserve the carbon and nitrogen despite the absence of this variable. Such practices would enhance mass conservation validation for both data users and ESM groups. Currently, the absence of certain variables (Fig. A1) leads to ambiguity from the perspective of data users: it is unclear whether these variables are not modelled or simply not reported.



**8.2 For C4MIP and related MIPs**

a) For the carbon cycle, collaborations between C4MIP, the Land Use Model Intercomparison Project (LUMIP) (Lawrence et al., 2016), the Fire Model Intercomparison Project (FireMIP) (Rabin et al., 2017), and possibly others should be considered to harmonize data requests and avoid overlap in variables with similar definitions. Currently, the coexistence of variables such as fLuc and fAnthDisturb, or fFire and fFireNat, not only places an additional burden on ESM teams for data reporting but also creates confusion for data users. From a user perspective, relying on model-specific variables to achieve mass conservation is far from ideal. Addressing this issue should be a priority in future updates to the MIP protocols.

b) It is suggested to promote fDeforestToAtmos and fHarvestToAtmos (Tier-2 fluxes for fAnthDiturb), fVegFire and fLitterFire (Tier-2 fluxes for fFireNat), and rhLitter and rhSoil (Tier-2 fluxes for rh) to Tier-1 variables as they are necessary for analysing mass conservation in key subpools.

c) In the nitrogen cycle, it is suggested to add a new Tier-1 variable to represent the net nitrogen flux between land and atmosphere (e.g., netAtmosLandNgasFlux, similar to nbp and netAtmosLandCO2Flux in the carbon cycle). If ESMs calculate this flux internally (thus, no data processing for CMIP data submission involved), it will aid ESM groups and data users in identifying whether any imbalance originates from fluxes or pool size discrepancies.

d) It is also suggested to promote fNgas and fNleach to Tier-1 variables and remove fNloss, considering their distinct destinations (atmosphere vs. ocean). This might be necessary for the future analyses on the land nitrogen cycle and its interaction with the atmosphere and ocean.

**8.3 For users of existing CMIP6 data**

a) It is recommended to prioritize pool size, 'net primary productivity' (npp), 'heterotrophic respiration' (rh), and 'net biome productivity' (nbp) data. For instance, in carbon cycle feedback calculations, utilizing consistent carbon pool size differencing, as exemplified in CMIP6 carbon cycle feedback analyses (Arora et al., 2020), while avoiding time-integrated fluxes used in CMIP5 and earlier studies (Arora et al., 2013; Friedlingstein et al., 2006), is advisable.

b) When a closed carbon-nitrogen cycle is strictly required, specifying how imbalance fluxes are calculated and managed becomes crucial. Data users may consider two main approaches for carbon mass conservation: (i) Directly using npp and rh flux data and manually calculating the imbalance flux, attributing it to disturbances (land use or fire emissions, which should not be assumed as zero in many models/experiments) or land-to-ocean carbon transport flux. (ii) Using npp, rh, fLuc, and fFire (where available) and manually calculating the imbalance flux, attributing it to other disturbances or land-to-ocean transport.



c) For nitrogen mass conservation, prioritizing the use of pool size data, 'biological nitrogen fixation' (fBNF, closely linked to npp), 'nitrogen deposition' (fNdep, verifiable through the input4MIPs dataset), and 'nitrogen uptake' (fNup) and 'net mineralization' (fNnetmin, both significant nitrogen fluxes) is recommended. Then, attributing imbalance fluxes in mineral and organic nitrogen pools to mineral nitrogen loss and anthropogenic perturbation, respectively, might be considered with minimal data adjustments.

## 9 Conclusions

The CMIP6 data archive serves as the primary data repository for various communities concerned with climate change. Building and maintaining such a vast data infrastructure is a collaborative effort, understandably fraught with challenges. In this study, we have identified a common issue of mass conservation in reported CMIP6 data from ESMs and experiments following the C4MIP protocol, resulting in accumulated mass imbalances reaching hundreds of gigatons of carbon/nitrogen. The potential reasons include incomplete data submission, inconsistent definitions of fluxes with the C4MIP protocol, different definitions of reporting, particularly concerning land use and fire emissions, and numerical errors from spatio-temporal regridding (minor). The presence of mass imbalance limits the usability of the data and may exacerbate uncertainties in results/conclusions derived from the data. Therefore, we recommend that ESM groups and the next generation of CMIP, C4MIP, and related MIPs prioritize mass conservation as a fundamental data validation step and document any identified mass imbalances. This would take effort, but we think this is a reasonable minimum step to justify the effort required to process, share and analyse the data included in CMIP. We also suggest that data users that require closed carbon/nitrogen cycles prioritize carbon/nitrogen pool size data and major carbon/nitrogen flux data (ignoring other data which would conflict with the mass balance inferred from these sources), while also specifying how they handle any remaining imbalance in their analyses (for example, attributing these imbalances to anthropogenic disturbances).

Achieving these goals requires collaborative efforts among the CMIP data request team, related MIPs, and modeling groups. Given the substantial efforts already invested in making ESM outputs publicly accessible and analyzable, conducting mass conservation verification is valuable. This step not only facilitates data analysis and reanalysis but also reduces uncertainty in the results and conclusions.





## Appendices

**Figure A1. Overview of data availability for the studied CMIP6 ESMs and experiments. The green dots indicate the data is available from ESGF.**

620

## Text A1. Methods for calculating the change in the size of the pools

The reported pool size data represents the mean value for each year, which provides no information about the size of the pool at the beginning or the end of the year. Discrete differencing of the pool size data using [pool size(t+1) - pool size(t)] actually calculates the difference between the midpoints of year (t+1) and year (t), rather than the within-year change. In 625 contrast, the reported flux data represents the average flux over the course of year (t). This discrepancy means that the pool size change derived from discrete differencing [pool size(t+1) - pool size(t)] does not correspond to the net flux derived from the reported flux data, potentially leading to a biased estimation of the net flux.





The following provides an example to illustrate the difference between gradient differencing and discrete differencing.
Assume a constant flux of 1 GtC/yr over ten years, with the pool size starting at zero. The true pool size time series would be [0, 1, 2, ..., 10] (eleven data points, unit: GtC). However, since the reported pool size is the mean value for each year, the pool size data would be [0.5, 1.5, 2.5, ..., 9.5] (ten data points, unit: GtC, Fig. A2). Discrete differencing of this pool size data yields a flux of 0.5 GtC/yr for the first year (pool size data at t=1 minus zero start, Fig. A2), which is lower than the actual flux of 1 GtC/yr. By applying gradient differencing, this issue can be avoided (Fig. A2).

The same principle can be applied to more complex examples. The point here is that this is a clear issue but it is also a small one, because we have relatively long time series and relatively well behaved data (so, while we don't have information about the exact values at the start and of the year, the other information provides enough information to constrain their possible values).

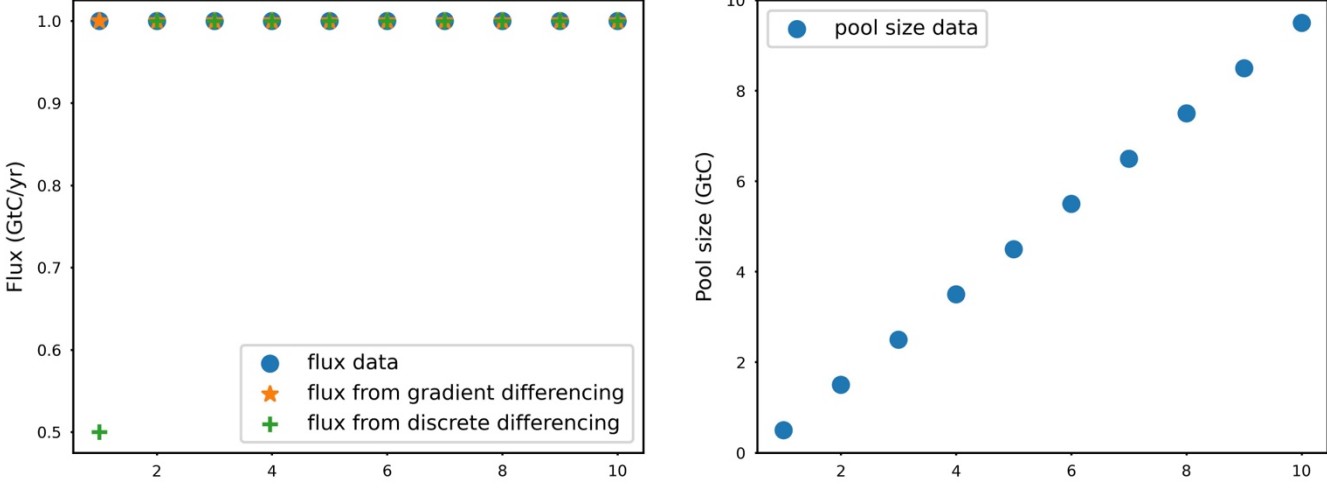

**Figure A2. The comparison of differencing methods for pool size data time series**

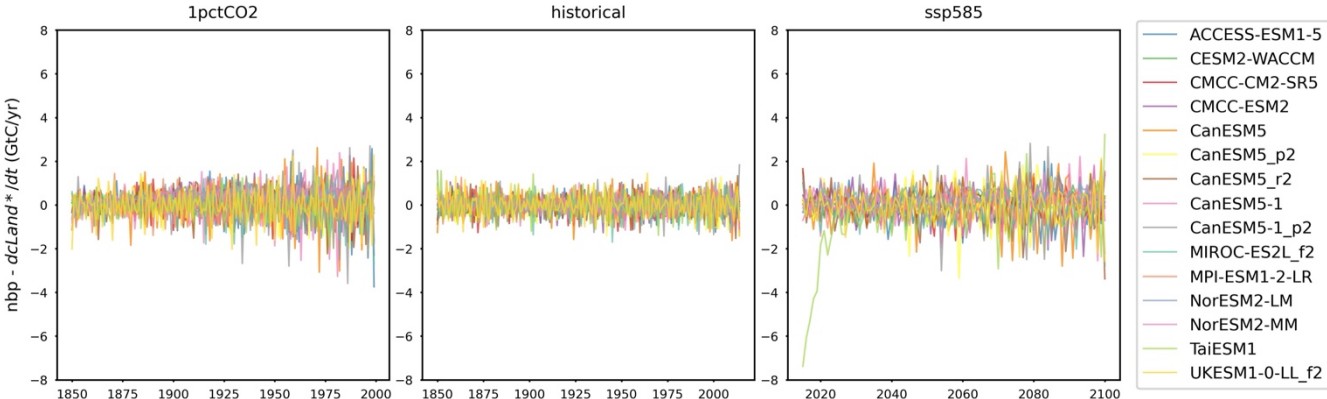

**Figure A3. The difference (imbalance flux) between 'net biosphere production' (nbp) and the change of land carbon pool size over time (dcLand*/dt, derived from pool size data) from CMIP6 ESMs across different scenarios**



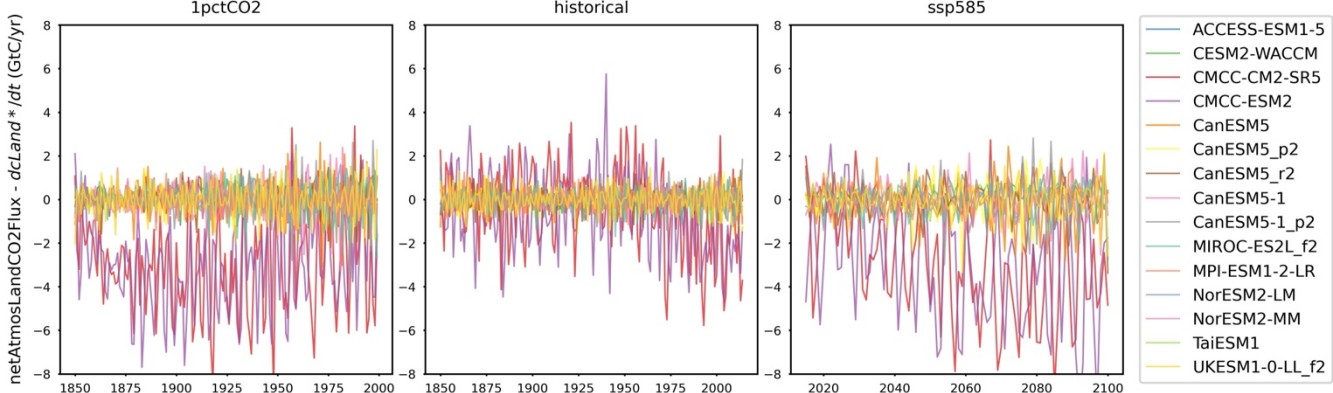

**Figure A4. The difference (imbalance flux) between netAtmosLandCO2Flux and the change of land carbon pool size over time (dcLand\*/dt, derived from pool size data) from CMIP6 ESMs across different scenarios**

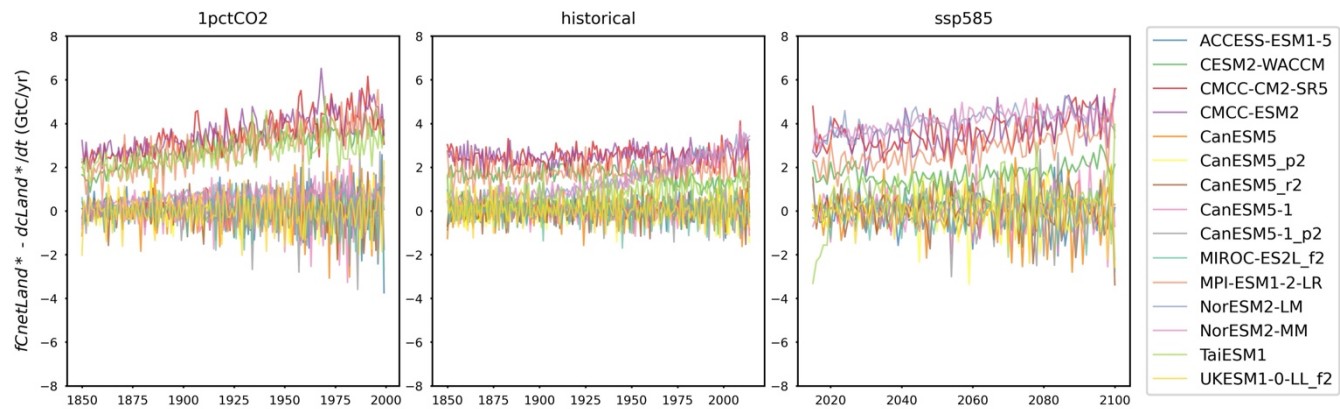

**Figure A5. The difference (imbalance flux) between 'net land carbon flux' (fCnetLand\*, reconstructed from flux data) and the change of land carbon pool size over time (dcLand\*/dt, derived from pool size data) from CMIP6 ESMs across different scenarios**



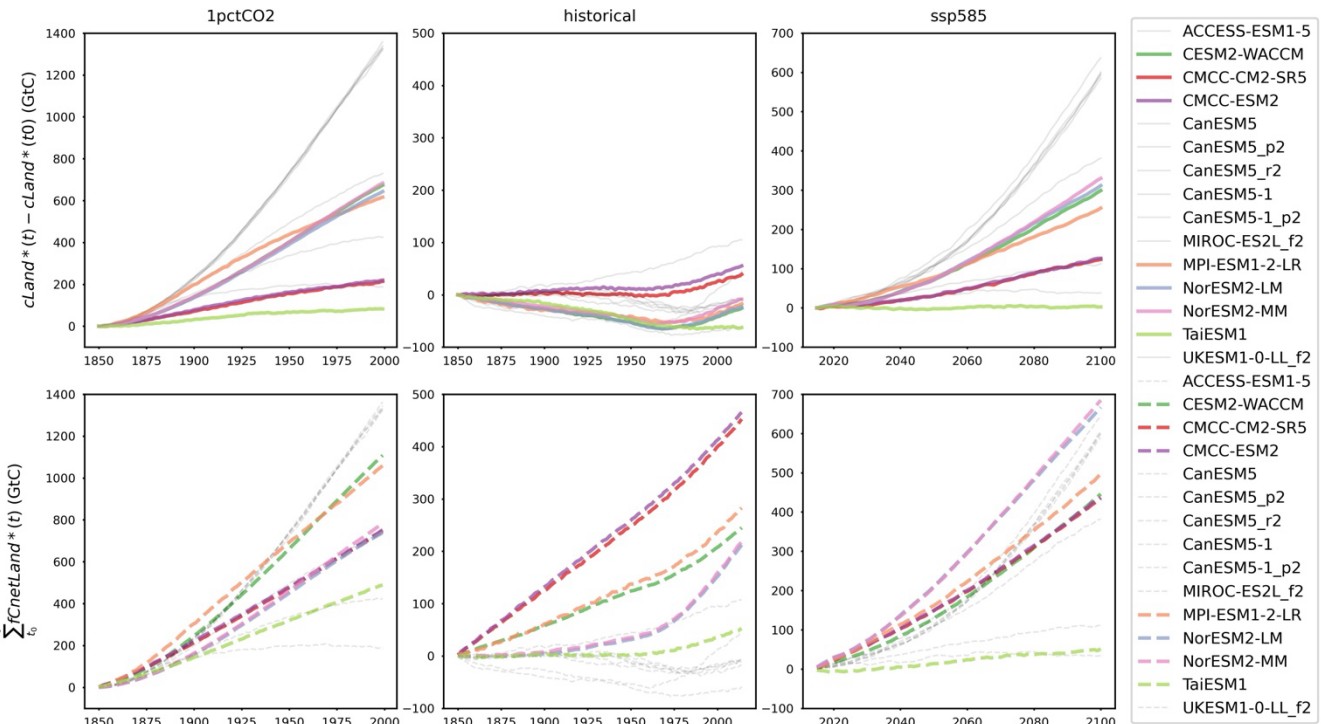

**Figure A6. The mismatch of changes in land carbon pool size derived from pool size data [cLand*(t) - cLand*(t0)] (solid lines) and those retrieved from the integration of net land carbon flux reconstructed from flux data (fCnetLand*, dashed lines) in CMIP6 ESMs across different scenarios. The carbon mass-conserved ESMs are all plotted with grey lines.**

**Text A2. The carbon mass imbalance from the land carbon subpools**

For the vegetation carbon pool, the reconstructed net flux (fCnetVeg*) consistently overestimates the vegetation carbon pool size change (dcVeg/dt) in all experiments for CESM2-WACCM, CMCC-CM2-SR5, CMCC-ESM2, MIROC-ES2L, and TaiESM1 (Fig. A7). The fCnetVeg* overestimates the pool size change in the historical and ssp585 runs of both NorESM models, while underestimating the change in the 1pctCO2 run. All CanESM models and UKESM1-0-LL show the matching fCnetVeg* and dcVeg/dt.

The mismatch between net flux and pool size change is less pronounced for the litter+soil carbon pool (fCnetLitterSoil* and dcLitterSoil/dt) compared to the vegetation pool (Fig. A7). Nonetheless, CESM2-WACCM, CMCC-CM2-SR5, and CMCC-ESM2 show a significant overestimation by the reconstructed net flux (fCnetLitterSoil* > dcLitterSoil/dt) in all experiments. The reconstructed fCnetLitterSoil* also overestimates the litter+soil pool size change in all experiments from NorESM2-LM and NorESM2-MM, though the historical run shows a smaller overestimation compared to their 1pctCO2 and ssp585 runs



(Fig. A7). All CanESM models, MIROC-ES2L, TaiESM1, and UKESM1-0-LL exhibit closer alignment between fCnetLitterSoil* and dcLitterSoil/dt, although cumulative imbalances remain substantial in certain cases (Fig. 4).

The product pool presents a smaller pool size change compared to the vegetation and litter+soil pools (Fig. A7, around zero where the product pool size is reported). The comparison of the reconstructed net flux (fCnetProduct*) and the derived pool size change over time (dcProduct/dt) shows a significant underestimation (fCnetProduct* < dcProduct/dt) for CESM2-WACCM and MIROC-ES2L. In CESM2-WACCM, this underestimation arises because the influxes (fDeforestToProduct and fHarvestToProduct) are much smaller than the outflux (fProductDecomp) - for example, 0.03 and 0.84 GtC/yr versus 3.6 GtC/yr during ssp585, on average. For MIROC-ES2L, the mismatch results from a non-zero fProductDecomp while no

fDeforestToProduct or fHarvestToProduct fluxes are reported (Fig. A1). ACCESS-ESM1-5, CMCC-CM2-SR5, CMCC-ESM2, MPI-ESM1-2-LR, and UKESM1-0-LL show nearly identical fCnetProduct* and dcProduct/dt in all experiments. Other ESMs do not report product carbon pool sizes but do report certain fluxes into or out of the product pool (Fig. A1), resulting in a non-zero net flux (Fig. A7).

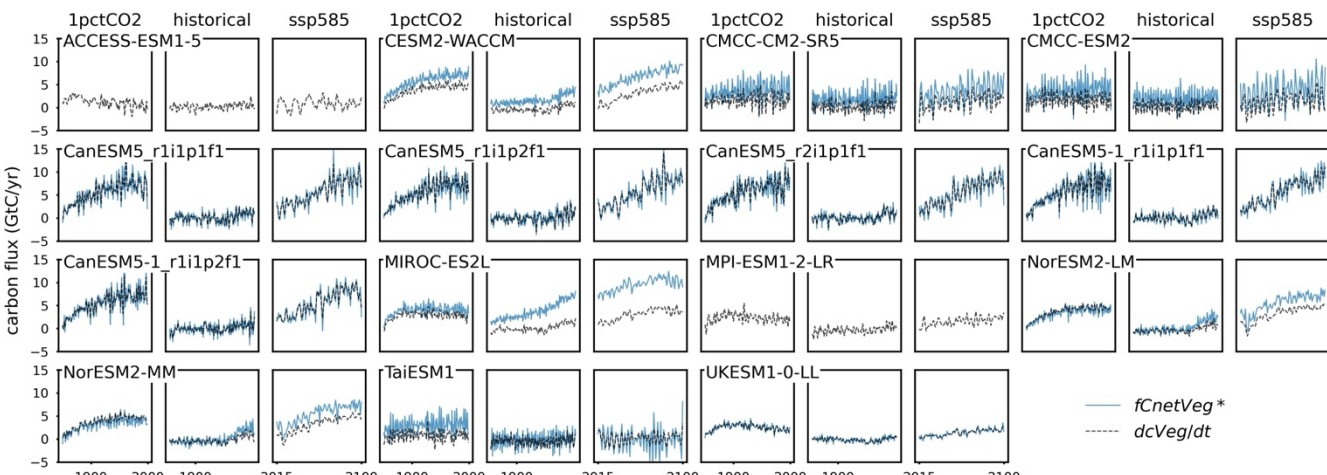



**Figure A7. Comparison of net carbon flux for vegetation, litter+soil, and product pools (fCnetVeg*, fCnetLitterSoil*, and fCnetProduct*, reconstructed from flux data) and the change of carbon pool size over time for vegetation, litter+soil, and product pools (dcVeg/dt, dcLitterSoil/dt, and dcProduct/dt, derived from pool size data) from CMIP6 ESMs across different scenarios.**







**Figure A8. Comparison of net nitrogen fluxes for organic and mineral pools (fNnetOrganic\* and fNnetMineral\*, reconstructed from flux data) and the change of nitrogen pool size over time for organic and mineral pools (dnOrganic\*/dt and dnMineral/dt, derived from pool size data) from CMIP6 ESMs across different scenarios.**





**Figure A9.** The mismatch of changes in land nitrogen pool size derived from pool size data [land nitrogen: nLand*(t) - nLand*(t0); organic nitrogen: nOrganic*(t) - nOrganic*(t0), and mineral nitrogen: nMineral(t) - nMineral(t0)] (solid lines) and those retrieved from the integration of net nitrogen flux reconstructed from flux data (land nitrogen: fNnetLand*, organic nitrogen: fNnetOrganic*, and mineral nitrogen: fNnetMineral*, dashed lines) in CMIP6 ESMs across different scenarios. The ESMs without a nitrogen cycle are all plotted with grey lines.






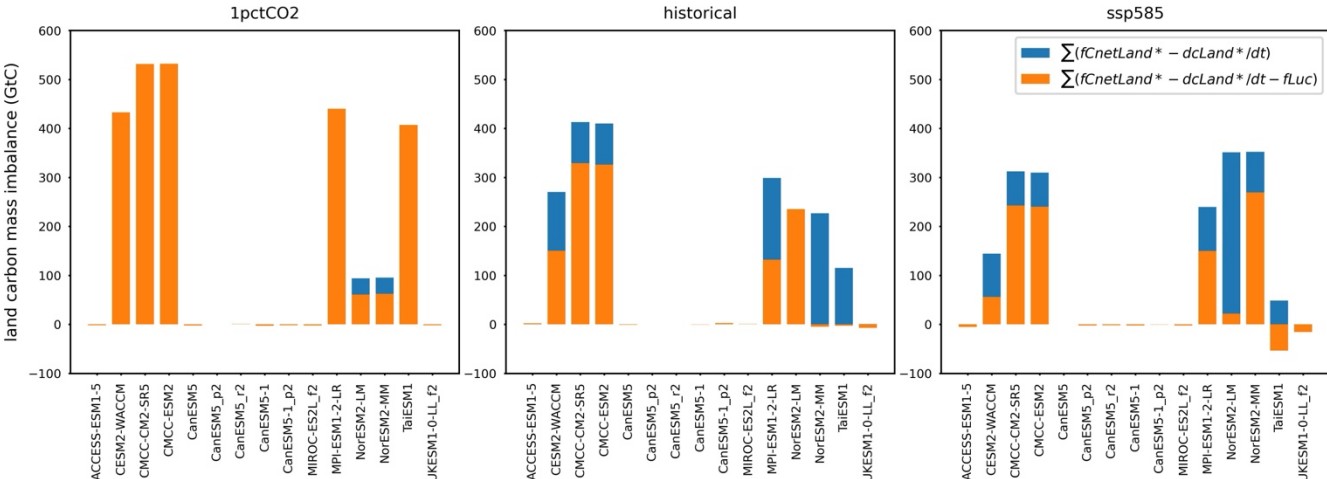

**Figure A10. The carbon mass imbalance from mismatch of net land carbon flux (fCnetLand\*, with or without the consideration of fLuc, reconstructed from flux data) and the change in land carbon pool size over time (dcLand\*/dt, derived from pool size data) for CMIP6 ESMs across different scenarios.**

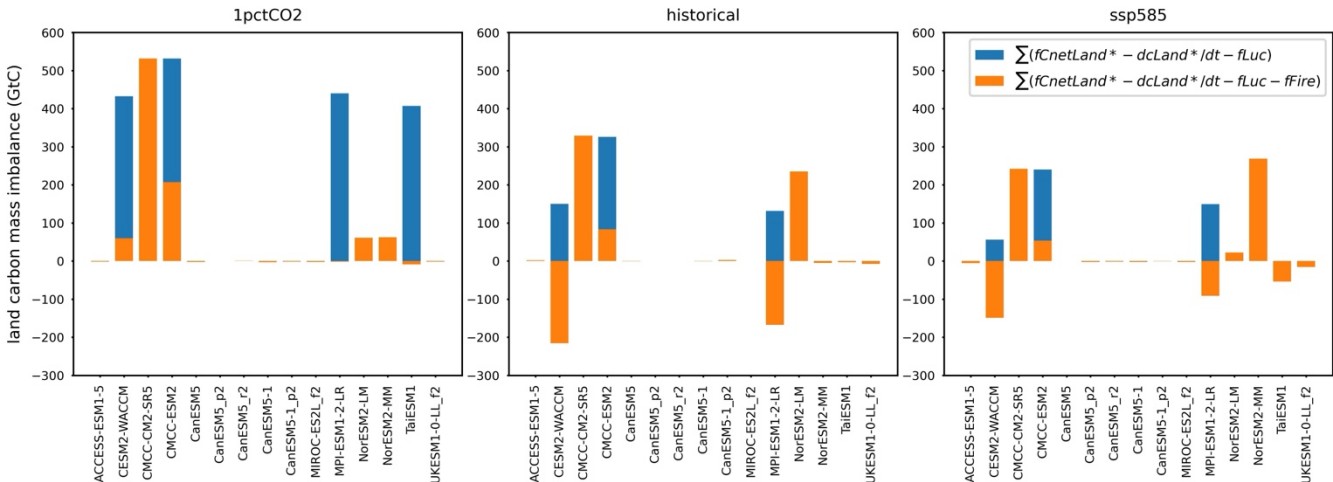

**Figure A11. The carbon mass imbalance from mismatch of net land carbon flux (fCnetLand\*, with or without the consideration of fFire, reconstructed from flux data) and the change in land carbon pool size over time (dcLand\*/dt, derived from pool size data) for CMIP6 ESMs across different scenarios.**



## Code and data availability

The CMIP6 Earth System Model (ESM) output data were initially collected from the Earth System Grid Federation (ESGF, https://esgf-node.llnl.gov/projects/cmip6/, last accessed on 25 June 2023). The processed global annual mean data, along with the associated calculation and visualization code, are available at https://doi.org/10.5281/zenodo.14060169.

## Author contribution

GT conceptualized the approach for analyzing carbon-nitrogen mass conservation in the published CMIP6 Earth System Model data. GT collected, processed, and analyzed the CMIP6 data. CJ suggested the analysis of mass conservation for
carbon and nitrogen subpools, which was subsequently conducted by GT. ZN, CJ, TG, TZ, AN, MM, and ARP contributed to the methodology and analysis, while MM, CJ, ZN, and AN supervised the study. GT analyzed and interpreted the results and prepared the first manuscript draft, with CJ, MM, ZN, and TG providing suggestions on structuring the narrative. All authors contributed to the writing and revision of the manuscript.

## Competing interests

One author is a member of the editorial board of Geoscientific Model Development (GMD).

## Acknowledgements

This work is supported by the National Environmental Science Program (NESP, Climate Systems Hub, Malte Meinshausen and Tilo Ziehn), funded by the Australian Government Department of Climate Change, Energy, the Environment, and Water; the Horizon 2020 Research and Innovation Funding Programme (No. 101003536, Earth System Models for the
Future, ESM2025, Zebedee Nicholls, Thomas Gasser, and Chris Jones), funded by the European Union; the Met Office Hadley Centre Climate Programme (Chris Jones), funded by the Government of the United Kingdom Department for Science, Innovation and Technology (DIST); and the Leeds-York-Hull Natural Environment Research Council (NERC) Doctoral Training Partnership (DTP) Panorama (No. NE/S007458/1, Alejandro Romero-Prieto).

We express our sincere gratitude to the World Climate Research Programme (WCRP), the Coupled Model Intercomparison Project Phase 6 (CMIP6), the Earth System Model (ESM) groups, the Coupled Climate-Carbon Cycle Model Intercomparison Project (C4MIP) and related MIPs, and the Earth System Grid Federation (ESGF) for their invaluable collaborative efforts in making ESM outputs available and accessible.



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
