# Peer review of "Investigating Carbon and Nitrogen Conservation in Reported CMIP6 Earth System Model Data"

_EGUsphere, 2024_

## Referee Comment (RC1)

**Investigating Carbon and Nitrogen Conservation in Reported CMIP6 Earth System Model Data**

**Gang Tang, Zebedee Nicholls, Chris Jones, Thomas Gasser, Alexander Norton, Tilo Ziehn, Alejandro Romero-Prieto, Malte Meinshausen**

The authors examine mass conservation in carbon and nitrogen data reported by multiple Earth System Models involved in CMIP6. Their analysis reveals substantial accumulated mass imbalances, potentially reaching hundreds of gigatons of carbon, which introduces significant uncertainty into the models' results and conclusions. The authors attribute these imbalances to missing fluxes in reported data and inconsistent definitions. Consequently, they recommend that future CMIP reporting protocols incorporate mass conservation checks in their validation processes, include secondary variables to facilitate mass balance calculations, and standardize definitions and variable names to reduce complexity. This topic is of crucial importance, as CMIP model results are widely used to understand the dynamics of numerous variables and their effects on climate change and variability. Moreover, when evaluating the diverse and sometimes contradictory results from various models, having additional data to assess model reliability would be invaluable in determining which models are more trustworthy. While the analysis and recommendations are highly relevant, the manuscript's presentation could be improved for better comprehension. To enhance readability, I recommend simplifying the narrative by eliminating repetitive paragraphs and reducing the use of parentheses. Additionally, it would be beneficial to define variables and scenarios clearly at the outset and streamline the manuscript's organization by reducing the number of subsections. These changes would make the document more accessible and easier to follow, facilitating a better understanding of the analyses ans results.

**Major comments:**
- I found the manuscript somewhat challenging to follow. I recommend adopting a more direct writing style and avoiding repetitive information. The presence of lengthy sentences and excessive parenthetical information complicates comprehension. Additionally, I believe the manuscript could be significantly shortened without losing essential content.
- I noticed an abundance of sections and subsections that could be streamlined by merging some of them. This would help eliminate repetitive content and enhance the manuscript's overall clarity and flow.
- The manuscript would be more accessible if all procedures and variable definitions were presented clearly from the outset. I found that the authors

provide information gradually, which adds unnecessary complexity to the manuscript.

**Minor comments:**
- Please replace CO2 with $CO_2$ throughout the manuscript.

Abstract
- Could the authors clarify what they mean by "Given that CMIP6 data is no longer being reported"?

**Introduction**
- L37. Please add a space before the references.
- L39-40. References are enclosed in double parentheses.
- L49. Please clarify the specific role being referenced and provide more detailed information about its significance.
- L75. What about sections 2 and 3? Please address these sections as well.
- L82. Consider removing the word "including" as there are no suggestions provided for other stakeholders.
- L85-93. Consider eliminating these reasons here, as they are addressed later in the text.

**Section 2**
- L101. Consider providing a brief description of the experiments.
- L103. Consider referencing a table that lists the names of the models instead of enumerating them here.
- L107. Consider clarifying what "variant_label" refers to.
- L112-114. Please rewrite this statement clearly.
- L118-119. What do the authors mean by "the monthly global pool sizes/fluxes were weighted using the model-specific calendar to calculate their annual mean"?

**Section 3**
- L129. What do the authors mean by "requires consideration". Please provide more specific details.
- L140. I am unclear about why the numerical errors are expected to be minimal based on the information provided in the previous sentence.

**Section 4**
- Eqs 2-10. Please provide definitions for each variable.
- L165. Please clarify the meaning of "The nbp and the dcLand*/dt exhibit the exact relationship as shown in Eq. 2."
- L166. There are not many details about this in Section 3, as mentioned.
- L186. Which results are you referring to? Please cross-reference the figures for each case.
- L189. Please provide a brief explanation of why it suggests that the differences are solely due to processing issues.

- L199. During which period?
- L210-211. Why does that indicate unavoidable numerical errors?
- L140. What do the authors mean by "slightly adjust the calculation of the equations mentioned earlier"? Additionally, please cross-reference those equations using their respective numbers instead of referring to them as "above."

**Section 5**
- L269. Why is it stated as "should have"?
- L280. Are there differences between the reported nLand and the calculated nLand*?
- L281. There is a typo in "Notably".
- L324. Which experimental periods are you referring to?
- L339. The value of the orders is missing.

**Section 6**
- Title. Consider using a more descriptive title.
- L379. Which composite variables should be removed?

**Section 7**
- Section 7.2. Consider highlighting the most relevant models, specifically those that demonstrate a "better conservation of mass."
- Table 1. Consider adding the values of imbalance, with distinctions indicating whether they are lower or greater than 5 Gt, along with the effect of the variable.
- L461. Why use the term "except"? Do you mean "besides"?
- L486. What kind of complexities are you referring to? Please be more specific.

**Section 8**
- L539-540. Consider rewriting this paragraph in a more friendly tone. In my opinion, as it stands, it seems there is little value in sharing the data.
- L544. Please add the reference for this claim.
- L66-666. I found this important; please consider including it in the main text.

**Figures**
- Fig. 1. This figure is frequently cited throughout the manuscript, but it is not adequately explained anywhere.
- Fig. 2. Why is "net biosphere production" placed in quotes in the legend?
- Fig 2. Consider plotting dcLand/dt in a different manner to facilitate easier differentiation.
- Figs. 5, 6, A7, A8, and A11. Please remove the CanESM models from these figures.
- Fig. A4. Please extend the y-axis to fully display the negative values.
- Fig. A6 and A9. Consider using the same colors and line types for the models in both the upper and lower panels, and include a consistent and unique legend for all graphs.

---

## Author Response (AR1)

**Author Comments (ACs)**

Tang, G., Nicholls, Z., Jones, C., Gasser, T., Norton, A., Ziehn, T., Romero-Prieto, A., and Meinshausen, M.: Investigating Carbon and Nitrogen Conservation in Reported CMIP6 Earth System Model Data, EGUsphere [preprint], https://doi.org/10.5194/egusphere-2024-3522, 2024.

In this Author Comments:

- The original referee comments are in black (directly copied from the comments).

- Our responses are in blue.

- *The text we quoted from the manuscript is in gray italics.*

We sincerely thank all referees for their constructive comments and feedback on our manuscript.

Best regards,

Gang Tang (GT, as referenced below)

on behalf of all co-authors

Top-level updates before addressing individual comments:

- The narrative of the paper:

  We acknowledge that both reviewers have raised comments regarding the narrative and/or structure of our manuscript. At this stage, we have not yet made changes to the overall structure. This is because we initially drafted the paper using an alternative narrative: introducing all the theoretical mass conservation equations together before presenting the results. This approach significantly reduced the number of subsections and made the flow more straightforward. Based on our understanding, this is what was suggested by the reviewers. However, we later opted for the submitted version, which introduces the mass conservation equations and results gradually. We believe this approach is more accessible to a broader audience who may not be very familiar with the variables, as it alleviates the need to remember all the equations at once. That said, this approach unavoidably results in more subsections and may appear repetitive when discussing different pools and subpools.

  To facilitate further feedback, we have provided our earlier draft for the reviewers to reference: https://drive.google.com/file/d/12uPiwthIoVwA2EoL5nIttVVaQ_I0x5hv/view?usp=sharing We are not requesting a full review of the draft, but we hope the reviewers can consider it to assess whether the original narrative (as presented in the draft) indeed better suits a broader audience. Specifically, we seek feedback on whether the "conservation equations presented together, followed by results" approach (in the earlier draft) or the "gradual introduction of conservation equations and results for different pools" approach (in the submitted verison) is more appropriate. As outlined in our responses to individual comments, we greatly appreciate any further feedback and are happy to make adjustments as needed. Thank you.

- Figure color scheme update:

  We have updated the color scheme for the line plots in our manuscript to ensure greater consistency and to make it more color-blind friendly, in line with the requirements of the GMD journal.

- A single PDF for all our responses to reviewers' comments:

  This Author Comments document provides our responses to all the reviewers' comments. As some comments are shared between both reviewers, we have addressed each comment individually and then combined our responses into a single PDF. This format allows reviewers to cross-check our responses to different questions.

**RC1: 'Comment on egusphere-2024-3522', Anonymous Referee #1, 20 Dec 2024**

The authors examine mass conservation in carbon and nitrogen data reported by multiple Earth System Models involved in CMIP6. Their analysis reveals substantial accumulated mass imbalances, potentially reaching hundreds of gigatons of carbon, which introduces significant uncertainty into the models' results and conclusions. The authors attribute these imbalances to missing fluxes in reported data and inconsistent definitions. Consequently, they recommend that future CMIP reporting protocols incorporate mass conservation checks in their validation processes, include secondary variables to facilitate mass balance calculations, and standardize definitions and variable names to reduce complexity. This topic is of crucial importance, as CMIP model results are widely used to understand the dynamics of numerous variables and their effects on climate change and variability. Moreover, when evaluating the diverse and sometimes contradictory results from various models, having additional data to assess model reliability would be invaluable in determining which models are more trustworthy. While the analysis and recommendations are highly relevant, the manuscript's presentation could be improved for better comprehension. To enhance readability, I recommend simplifying the narrative by eliminating repetitive paragraphs and reducing the use of parentheses. Additionally, it would be beneficial to define variables and scenarios clearly at the outset and streamline the manuscript's organization by reducing the number of subsections. These changes would make the document more accessible and easier to follow, facilitating a better understanding of the analyses ans results.

Major comments:

I found the manuscript somewhat challenging to follow. I recommend adopting a more direct writing style and avoiding repetitive information. The presence of lengthy sentences and excessive parenthetical information complicates comprehension. Additionally, I believe the manuscript could be significantly shortened without losing essential content.

I noticed an abundance of sections and subsections that could be streamlined by merging some of them. This would help eliminate repetitive content and enhance the manuscript's overall clarity and flow.

The manuscript would be more accessible if all procedures and variable definitions were presented clearly from the outset. I found that the authors provide information gradually, which adds unnecessary complexity to the manuscript.

GT: Thank you very much for reviewing our paper and providing such insightful comments. We appreciate your feedback on the narrative and structure of our manuscript. The suggestions you made are very valuable. However, I would like to provide some background on the current version of the manuscript.

Initially, I started drafting the paper by introducing the theoretical mass conservation equations (for carbon and nitrogen across all pools and subpools), followed by the presentation of the results, and then the discussion, suggestions, and conclusions. However, the section on equations and their explanations became quite lengthy, and our co-authors were concerned that presenting all the equations in one place might overwhelm readers who are not familiar with these variables - it could be challenging to keep track of approximately 20 equations while reading the results. For reference, here is our earlier draft:

https://drive.google.com/file/d/12uPiwthIoVwA2EoL5nlttVVaQ_I0x5hv/view?usp=sharing

As a result, we moved to the current structure in the submitted version, where we first present the carbon conservation equation for the total land, followed by the results for the land, then repeat this for the carbon subpools and nitrogen conservation. We recognize that this approach has its disadvantages, as you pointed out - such as some repetitive descriptions (e.g., models not conserving land carbon or vegetation/litter/soil carbon), numerous subsections, and information being introduced gradually. However, we believe this structure helps reduce the cognitive load for the reader. It allows readers to focus on a few

key variables and one or two equations at a time, rather than having to remember them all or refer back to the equation section constantly. This is the main reason we chose the current structure for the submitted manuscript.

Personally, I do not have a strong preference for either option, as I am very familiar with all the carbon-nitrogen cycle variables in CMIP. We are also open to revising the narrative to consolidate all the equations and results in one place. Before making changes to the structure, we would like to ask if you could review our previous draft briefly to compare it with the current version. This is not a request to review the previous draft in full, but simply to help us gauge which narrative might be more accessible to a broader audience. If after your comparison you still feel that consolidating all the equations and results in one place is the better approach, please let us know. We would greatly appreciate any further comments and will be happy to make adjustments accordingly.

Minor comments:

Please replace CO2 with $CO_2$ throughout the manuscript. Abstract

GT: Thanks for the comment. We have now revised it to $CO_2$ in the manuscript.

Abstract

Could the authors clarify what they mean by "Given that CMIP6 data is no longer being reported"?

GT: Apologies for the confusion. What we meant is: "*Given that no additional CMIP6 data is currently being published and none is expected in the future.*" This has now been clarified in the manuscript.

Introduction

L37. Please add a space before the references.

GT: Revised, thanks.

L39-40. References are enclosed in double parentheses.

GT: Revised, thanks.

L49. Please clarify the specific role being referenced and provide more detailed information about its significance.

GT: Thanks for the comment. We have now added the details for the references.

*For example, the carbon-climate feedback can become negative when the nitrogen cycle is incorporated into models (Zaehle and Dalmonech, 2011; Zaehle et al., 2015). Observational and modeling studies suggest that nitrogen limitation reduces net primary productivity, a key carbon flux that removes $CO_2$ from the atmosphere (Lebauer and Treseder, 2008; Schulte-Uebbing and De Vries, 2018).*

L75. What about sections 2 and 3? Please address these sections as well.

GT: Thanks for the comment. We have now added the information about Seciton 2 and 3, which is quoted below:

*Sections 2 and 3 outline our data collection and processing approach, as well as the method for validating mass conservation.*

L82. Consider removing the word "including" as there are no suggestions provided for other stakeholders.

GT: Thanks for the comment. It has been remove. The revision is quoted below:

*Based on the results, we provide suggestions in Section 8 for the CMIP data request team, ESM groups, the C4MIP and related MIPs, and CMIP6 data users.*

L85-93. Consider eliminating these reasons here, as they are addressed later in the text.

GT: Thanks for the comment. It has been removed now.

Section 2

L101. Consider providing a brief description of the experiments.

GT: Thank you for the suggestion. We have now added Table A1, which provides detailed descriptions of the variables, models, and experiments.

L103. Consider referencing a table that lists the names of the models instead of enumerating them here.

GT: Thank you for the suggestion. It now refers to Table A1.

L107. Consider clarifying what "variant_label" refers to.

GT: Thank you for the suggestion. It is now explained in Table A1.

L112-114. Please rewrite this statement clearly.

GT: Thank you for the comment. We have updated this paragraph as follows:

*The C4MIP variables and related data (Fig. 1 and Table A1) were obtained from ESGF (https://esgf.llnl.gov/, last access 25 June 2023; data availability varies across different ESMs and experiments, Fig. A1). To ensure consistent data post-processing, the most frequently used monthly data with the model's native grid (with "gn" as "grid_label", see CMIP6 global attributes details at https://goo.gl/v1drZl, last access 25 June 2023) was collected.*

L118-119. What do the authors mean by "the monthly global pool sizes/fluxes were weighted using the model-specific calendar to calculate their annual mean"?

GT: Different models use different calendars for their outputs, meaning that the monthly data represent pool sizes or fluxes over varying numbers of days. We account for these differences when calculating the annual mean. For example, if a model uses a Gregorian calendar without leap years, and outputs monthly data such as cVeg_January (31 days) = 500 GtC, cVeg_February (28 days) = 510 GtC, …, the annual mean is calculated as: cVeg_annual_mean = (500 * 31 + 510 * 28 + … ) / (31 + 28 + …).

We have slightly revised the description (quoted below) to make it clear.

*To calculate their annual means, we accounted for differences in model-specific calendars by weighting the monthly global pool sizes/fluxes according to the number of days in each month.*

Section 3

L129. What do the authors mean by "requires consideration". Please provide more specific details.

GT: Thanks for the comment. This means that the second calculation, 'the change in the pool size over time,' is not straightforward and requires careful consideration. We have provided additional details in the following sentences (till the end of this paragraph), as quoted below:

*Both the pool size data and flux data are temporal mean values (a requirement from the CMIP6 data request2). However, this reporting convention means that there is only one pool size in each month, rather than information about the size of the pool at the beginning and end of the month (which is what you would need to unambiguously calculate the change in the pool size over the month of interest). As a result, blindly calculating the difference in pool size between each reported data point will inevitably result in a net flux that differs from the flux reported in the time series. Considering this, we employed a gradient method (Fornberg, 1988), which calculates gradients using second order accurate central differences in the interior points and first order accurate one-sides differences at the boundaries, to derive the change in the pool size from the reported pool size data. It should be noted that this approach is simply a method to obtain a better idea of the actual differential (example provided in Text A1 and Fig. A2). It does not fundamentally resolve the discrepancy between differencing discrete data and differentiating continuous data. Having said this, the numerical errors introduced due to the reporting convention and the method used to calculate the change in pool size are expected to be minimal.*

L140. I am unclear about why the numerical errors are expected to be minimal based on the information provided in the previous sentence.

GT: Thank you for the question. Although the data is reported as a monthly mean (a single value) rather than as values for the start and end of the month (two values), we can still estimate the actual differential using a time series of data. For example, if cVeg_January = 500 GtC and cVeg_February = 510 GtC, the monthly difference (from mid-January to mid-February) is +10 GtC. Extending this to a time window from start-January to end-January would yield a similar value, assuming no significant disturbances (which is reasonable when discussing global carbon pool sizes).

For a relatively long time series (e.g., 165 years of monthly data for the historical period), the numerical error introduced by calculating dcVeg/dt from discrete cVeg data can be minimized by differencing the series. While the time windows may differ slightly, the magnitudes of dcVeg/dt should be comparable. The gradient method we applied further reduces numerical error by addressing the differences in time windows. This method effectively extends the mid-month data to approximate start-month and end-month values, leveraging the full series of mid-month data to refine the calculation.

We have now included this justification in the manuscript, as quoted below:

*However, given the relatively long time series of monthly data (e.g., 165 years for the historical period), the numerical errors introduced by the reporting convention and the method used to calculate changes in pool size are expected to be minimal. In other words, while the discrepancy persists, the magnitudes of dy/dt derived from both methods should closely align.*

Section 4

Eqs 2-10. Please provide definitions for each variable.

GT: Thanks for the comment. We have now added Table A1 for the variable description.

L165. Please clarify the meaning of "The nbp and the dcLand*/dt exhibit the exact relationship as shown in Eq. 2."

GT: Thanks for the comment. At the very beginning I wrote Eq. 2 as

  dcLand / dt = netAtmosLandCO2Flux (or nbp) - fCLandToOcean.

However, including '(or nbp)' within the equation looked awkward. Therefore, we decided to remove it and added a note stating:

*b) netAtmosLandCO2Flux and nbp (a CMIP5 variable) are interchangeable based on their definitions (which may also be part of the confusion, see details in Section 6.2);*

Unfortunately, we overlooked revising the corresponding text in the results section, leading to confusion. We apologize for this oversight. The description has now been revised as follows:

*The nbp and dcLand*/dt exhibit nearly identical patterns across all the studied ESMs and experiments (Fig. 2 and Fig. A3), with the small differences likely resulting from the differentiation method applied (see details in Section 3).*

L166. There are not many details about this in Section 3, as mentioned.

GT: Thank you for the comment. Please see our responses to the comments on L129 and L140 above. The revised Section 3 should now clearly explain the numerical differences.

L186. Which results are you referring to? Please cross-reference the figures for each case.

GT: Thank you for the comment. The results are shown in Fig. 2, which is referenced at the end of the sentence. To avoid the confusion, we have now revised the sentence as below:

*The reconstructed fCnetLand* and dcLand*/dt are well-aligned in ACCESS-ESM1-5, CanESM5-1, CanESM5, MIROC-ES2L, and UKESM1-0-LL (Fig. 2), with the difference fluctuating around zero and showing no discernible trends (Fig. A5),*

L189. Please provide a brief explanation of why it suggests that the differences are solely due to processing issues.

GT: Thank you for the comment. In our manuscript, we wrote, "*with the difference fluctuating around zero and showing no discernible trends, suggesting that these differences are due to numerical processing issues alone (Fig. A5).*" The reasoning is that if a clear trend was present (as seen in the other models in Fig. A5), it would be more likely due to a missing flux or a flux with incorrect values.

L199. During which period?

GT: Thanks for the comment. The 60 GtC represents the amount of imbalance accumulated over the entire 1pctCO2 experimental period.

L210-211. Why does that indicate unavoidable numerical errors?

GT: Thanks for the comment. Please refer to the above response to the comments on L129 and L140.

L140. What do the authors mean by "slightly adjust the calculation of the equations mentioned earlier"? Additionally, please cross-reference those equations using their respective numbers instead of referring to them as "above."

GT: Details about the adjusted calculation are provided in Section 6.4, as it is a relatively minor issue (quoted below). Specific equations are referenced in the explanation.

*For carbon conservation in vegetation and litter-soil pools, consideration of the Tier-2 component fluxes (fVegFire and fLitterFire) for fFireNat is unavoidable (Eq. 5 and Eq. 7). However, fFireNat itself is reported only in TaiESM1 and the two NorESM models (Fig. A1), and component fluxes are even rarer (only CESM2-WACCM reports fVegFire, and none of the studied models report fLitterFire, Fig. A1). Consequently, in calculating fCnetVeg\* and fCnetLitterSoil, we replace fVegFire in Eq. 5 with fFireNat and omit fLitterFire from Eq. 7. While this adjustment is not ideal, it is the only feasible approach given the limited data availability.*

Section 5

L269. Why is it stated as "should have"?

GT: Thanks. Because this is theoretical mass conservation.

L280. Are there differences between the reported nLand and the calculated nLand\*?

GT: Yes, we have now included the details in Section 6.1, as quoted below:

*In many of the ESMs examined in this study, the reported total land carbon or nitrogen (cLand or nLand) does not match the sum of the respective subpools (i.e., the reconstructed cLand\* or nLand\*). The discrepancy averages between 50 and 150 GtC or 25 and 55 GtN across the three experimental periods.*

L281. There is a typo in "Notably".

GT: Revised. Thanks for checking.

L324. Which experimental periods are you referring to?

GT: Thanks for the comment. "*MIROC-ES2L, MPI-ESM1-2-LR, NorESM2-LM, and NorESM2-MM are relatively mass-conserved in their total land nitrogen pool (mass imbalance <5 GtN, particularly in their 1pctCO2 runs (mass imbalance <0.8 GtN).*" The first part of the sentence we mean for all the experiments.

L339. The value of the orders is missing.

GT: Thanks for the comment. We did not specify the value here because we wanted to convey that the difference is large, but it may vary across different models (i.e., the difference could range from one to three orders of magnitude).

Section 6

Title. Consider using a more descriptive title.

GT: Thanks for the suggestion. We have now changed the title to "*6 Other issues with the reported CMIP6 data*"

L379. Which composite variables should be removed?

GT: Thank you for the comment. This is a general comment. Since the composite flux can be reconstructed from its component fluxes, it makes sense to prioritize the component fluxes and remove the composite fluxes (from our perspective). However, there may be reasons for the existence of composite fluxes and their initial creation that we are not aware of. Therefore, we have avoided specifying which variables we suggest prioritizing or removing.

Section 7

Section 7.2. Consider highlighting the most relevant models, specifically those that demonstrate a "better conservation of mass."

GT: Thank you for the comment. The "Y" and "N" (yes and no) in Table 1 directly and clearly indicate whether the data from models conserve mass.

Table 1. Consider adding the values of imbalance, with distinctions indicating whether they are lower or greater than 5 Gt, along with the effect of the variable.

GT: Thanks for the suggestions. The notes accompanying Table 1 (quoted below) explain the meaning of the signs, including the magnitude of the mass imbalance and the impact of the added variables.

*Y: mass conserved (cumulative imbalance <= 5GtC)*

*N: mass not conserved (cumulative imbalance > 5GtC)*

*N (-): mass not conserved, but including this variable decreased mass imbalance (make things better)*

*N (+): mass not conserved, and including this variable increased mass imbalance (make things worse)*

*N (n.d.): mass not conserved, no data for this variable*

*/: mass conserved with C4MIP variables so no further processing*

L461. Why use the term "except"? Do you mean "besides"?

GT: Thanks. We have revised the wording accordingly.

L486. What kind of complexities are you referring to? Please be more specific.

GT: Thank you for the comment. Here, we refer to the nitrogen cycle as encompassing both organic and mineral nitrogen, which involves additional fluxes and processes. Both components must adhere to mass conservation, making nitrogen mass conservation more challenging. We have now clarified this in the manuscript, as quoted below:

*The conservation of land nitrogen mass presents a more intricate challenge due to the complexities inherent in the nitrogen cycle (e.g., the need to conserve both organic and mineral nitrogen, Fig. 1).*

Section 8

L539-540. Consider rewriting this paragraph in a more friendly tone. In my opinion, as it stands, it seems there is little value in sharing the data.

GT: Thank you for the comment. We certainly value the sharing of data, but we aim to emphasize the importance of basic validation checks to ensure data quality before making it widely accessible. Now we have revised the paragraph accordingly to better reflect this viewpoint, as quoted below:

*Given the significant effort required to manage and share large data volumes, it is reasonable to prioritize these fundamental checks to guarantee that the data shared is both reliable and valuable to the broader community.*

L544. Please add the reference for this claim.

GT: Thanks for the comment. The statement "*Currently, a disconnect exists between these two - ESMs often find it challenging to fully align with MIP protocols, leading to incomplete data reporting or reported fluxes that do not adhere to the prescribed definitions (as evidenced by the fLuc and fFire).*" reflects our understanding of why issues like incomplete data reporting and inconsistencies in the definitions of requested variables occur. To clarify our point, we have now revised it as follows:

*Currently, incomplete data reporting (Fig. A1) or fluxes that do not adhere to the prescribed definitions (as evidenced by the fLuc and fFire) indicate a disconnect between ESMs and MIP protocols, as ESMs may find it challenging to fully align with these protocols.*

L66-666. I found this important; please consider including it in the main text.

GT: Thanks for the suggestion. We have now added the key information "*Notably, all CanESM models and UKESM1-0-LL show conserved carbon in the vegetation pool.*" into the main text.

Figures

Fig. 1. This figure is frequently cited throughout the manuscript, but it is not adequately explained anywhere.

GT: Thank you for the comments. Figure 1 is directly adapted from the C4MIP protocol (with the original paper cited). It illustrates all the fluxes (indicated by arrows with their respective directions) and pools requested by C4MIP. We did not provide all the details in one section because the mass conservation principles apply to different parts of the figure, depending on the pool in question. Consequently, the figure is referenced in multiple sections, with the relevant details discussed alongside the corresponding content. To address your concern, we have now added more explanation to the figure caption:

*Figure 1. Land carbon-nitrogen pools and fluxes requested by the C4MIP protocol. The boxes represent pools, while the arrows indicate fluxes and their directions. Details of variable descriptions are provided in Table A1. This figure is reproduced from the C4MIP protocol (Jones et al., 2016) under the Creative Commons Attribution 3.0 License.*

Fig. 2. Why is "net biosphere production" placed in quotes in the legend?

GT: Thank you for the comment. For all figure captions, we consistently use the format 'xxx' (abbr.) to introduce abbreviations. This approach enhances the readability of each figure, allowing readers to understand it without needing to refer to the full paper.

Fig 2. Consider plotting dcLand/dt in a different manner to facilitate easier differentiation.

GT: Thank you for the suggestion. Here we aim to highlight the matching between dcLand*/dt and nbp (or netAtmosLandCO2Flux). The black dashed line is used to prevent overlapping of the colored lines.

Figs. 5, 6, A7, A8, and A11. Please remove the CanESM models from these figures.

GT: Thanks we have updated these figures.

Fig. A4. Please extend the y-axis to fully display the negative values.

GT: Thanks we have updated the figure A4.

[Figure]

Fig. A6 and A9. Consider using the same colors and line types for the models in both the upper and lower panels, and include a consistent and unique legend for all graphs.

GT: Thanks we have revised the figures accordingly.

**RC2: 'Comment on egusphere-2024-3522', Anonymous Referee #2, 16 Jan 2025**

Review on 'Investigating Carbon and Nitrogen Conservation in Reported CMIP6 Earth System Model Data'

First of all, apologies for the delay in the review.

The paper investigates the carbon and nitrogen mass balance in simulations from the Coupled Model Intercomparison Project Phase 6 (CMIP6) with the published flux and pool variables. The authors find a significant mass imbalance in the reported data for both carbon and nitrogen in various experiments. They discuss the possible causes of this imbalance, arriving at the conclusion that it is likely due to missing fluxes and inconsistency between the reported data and the variable definitions. This leads to the conclusion that for the upcoming CMIP requests, more consideration should be given to consistency of variable requests between MIPs and incorporating mass conservation into the data validation process.

The analysis in the paper is well done and highly relevant considering the upcoming CMIP7 data request. However, the writing of the paper could be improved. There is an excessive use of both lists and parenthesis in the main text, hindering the flow of the text. At times the authors seem to become defensive and scared about not wanting to offend other people through misconceptions. This is most prevalent in the disclaimer found in the last paragraph of the introduction. A scientific paper should just focus on facts without unnecessary disclaimers, defenses or attacks. The authors should furthermore revise the paper regarding sub- and superscripts in formulas and variable names. Especially the long variable names would benefit from proper subscripting for more readability. It would also be nice to add a table for the variables with the variable name, the long name or description, as well as its source, such as which CMIP data request it comes from. This table could then be introduced in section 2 and later referenced without needing to introduce variables and their definitions over time. The footnotes found in the paper should be turned into references to be cited.

With some adjustments in the writing, this paper will become a valuable resource for the CMIP carbon community.

GT: Thank you for reviewing our paper and providing such insightful comments. We have added Table A1 to consolidate the details about variables, models, and experiments, avoiding the need to list them throughout the manuscript. Additionally, the disclaimer has been removed, as it repeats information presented later in the paper. However, we believe that in most cases, the term "mass conservation issue" is naturally interpreted as referring to model physics rather than output data, which is not the intended meaning in our study. We have clarified this point in the manuscript.

Minor comments:

* denotes where my comments overlap with those of referee #1

line 23ff: 3 uses of 'the reported data' in one sentences seems excessive, consider rewording

GT: Thank you for your comment. We have revised it as follows:

*However, we postulate that the carbon mass imbalance primarily arises from missing fluxes in the reported data and inconsistencies between these data and the definitions provided by the C4MIP protocol (e.g., land use and fire emissions)*

*line 30: 'Given that CMIP6 data is no longer being reported, ...' - reported can be misleading to readers and unclear. Do you mean that errata are no longer reported? No new data is published? Errata published but error not fixed? Please rephrase.

GT: Thank you for the comment. What we mean is that no new data will be published for CMIP6. We have revised the text to clarify this.

*Given that no additional CMIP6 data is currently being published and none is expected in the future*

line 34: 'The Coupled Model Intercomparison Project (Phase 6 - the latest version, CMIP6)' - mentioning Phase 6 in the parenthesis while not referring to it outside

GT: Thank you for the comment. We have now revised it.

*The Coupled Model Intercomparison Project Phase 6 (CMIP6, the latest phase at the time of writing)*

*line 36 'climate(Meehl …' - missing space before parenthesis

GT: Revised. Thanks for checking.

*line 39f: '(such as (Nicholls et al., 2022; Turnock et al., 2020; Stouffer et al., 2017))' - wrong citation style while within the brackets

GT: Revised. Thanks for checking.

*line 47 (and others): 'CO2' - please subscript the 2 for $CO_2$. This occurs throughout the paper.

GT: Sorry we missed that. Now it has been checked and revised.

*line 74: no description given for sections 2 and 3.

GT: Thank you for the comment. Now we have added the description.

*Sections 2 and 3 outline our data collection and processing approach, as well as the method for validating mass conservation.*

line 80: 'relevant stakeholders, including ...' stakeholders seems an add choice of word in this context.

GT: Thanks for the comment. We have now revised the sentence as follows:

*Based on the results, we provide suggestions in Section 8 for the CMIP data request team, ESM groups, the C4MIP and related MIPs, and CMIP6 data users.*

*line 83ff: This paragraph reads overly defensive, while nothing written so far would suggest that you believe the mass balance to stem from the models themselves. The list of possible factors is also repeated

in the analysis in further chapters and does not belong to an introduction. I would suggest to remove this paragraph as it just duplicates what is found in later chapters.

GT: Thanks for the comment. We have now removed it.

*line 103ff: Instead of listing the models separately, you could refer to Figure A1.

GT: Thanks for the suggestion. Now we have added a Table A1 for all the details of variables, models, and experiments.

line 146: ' - one being greater than the other or vice versa' - I doubt it is necessary to explain what an over- or underestimation means in this context, this part of the sentence can be removed.

GT: Thanks for the suggestion. We have removed it and revised the sentence to make it clear.

*The terms "overestimation" and "underestimation" in this paper specifically denote cases where the net flux exceeds or is smaller than the pool size change in the reported data, rather than implying a comparison with observational data or any external benchmark.*

line 154: 'nbp (a CMIP5 variable)' - I believe nbp is also in the CMIP6 data request, so do you mean 'introduced in CMIP5'?

GT: Yes here we mean nbp is introduced in CMIP5. We have now revised it. Thanks.

line 199ff: As long as you are talking about the same figure, it is unnecessary to end every sentence with (Fig. 3). This should be done when starting to talk about another figure or referring to another figure later on. This also occurs later on for other figures.

GT: Thank you for the comment. We have reviewed the manuscript and removed unnecessary figure references where the same figure was mentioned multiple times.

line 235ff: 'fCnetLitterSoil ∗=' etc - please check all of your sub- and superscripting. This asterisk should be on the variable with a space before the equals sign, same with other variables. '∗=' has another meaning in programming languages and would be confusing.

GT: Thanks for checking. Now they are corrected.

*line 291: 'Notebally' - Notably

GT: Thanks and typo corrected.

line 386: '(time steps also vary within ESMs' sub-modules)' - unnecessary information at this point, can be removed

GT: Thanks and removed.

line 533ff: ' ESGF may develop necessary data quality control tools considering mass conservation for data publication.' - may need to develop

GT: Thanks and revised.

*line 537ff: 'At first, this may seem a difficult task to meet. However, we would ask a different question. What is the value of sharing data which does not pass the basic test of conserving mass?' - Doesn't seem to fit the writing style of this paper. I would suggest to remove these sentences and add the next sentence ('Given … ecosystem') on the previous paragraph, as it already contains everything you want to convey.

GT: Thank you for the suggestion. This is an excellent solution to concisely convey our intended message. We have incorporated the suggestion and revised the manuscript accordingly.

line 545: 'by the fLuc and fFire' - either drop the 'the' or include it for both

GT: Thanks for checking. We have removed both "the".

line 591: ', and 'nitrogen uptake' (fNup) and 'net mineralization'' - remove the first 'and'

GT: Thanks for checking. We have removed it.

line 599: 'gigatons of carbon/nitrogen' - 'carbon and nitrogen'

GT: Thanks for checking. We have revised it.

line 605: 'This would take effort, but ..' - Remove

GT: Thanks and removed.

line 621ff: 'Text A1' etc – Regular sections are also not named 'Text 1', so remove the 'Text' from the appendix section headings.

GT: Thanks for checking. Now we have moved the "Text".

*Figures 5, 6, A8, A9: As you don't have any nitrogen data from the CanESM models, you should remove them from the figures and the legends, and mention this at the beginning of section 5.

GT: Thank you for the comment. We have updated the figures and clarified that CanESM does not include a nitrogen cycle by adding "*Besides, the CanESM series models do not include a nitrogen cycle and are therefore excluded from the following figures.*" at the beginning of nitrogen conservation result section.